# Do Current Large Language Models Master Adequate Clinical Knowledge?

## Abstract

Large Language Models (LLMs) show promising potential in solving clinical problems. Current LLMs, including so-called medical LLMs, are reported to achieve excellent performance on certain medical evaluation benchmarks, such as medical question answering, medical exams, etc. However, such evaluations cannot assess whether LLMs have mastered sufficient, compressive, and necessary medical knowledge for solving real clinical problems, such as clinical diagnostic assistance. In this paper, we propose a framework to assess the mastery of LLMs in clinical knowledge. Firstly, we construct a large medical disease-based knowledge base, MedDisK, covering 10,632 common diseases across 18 clinical knowledge aspects, which are crucial for diagnosing and treating diseases. Built on that, we propose a MedDisK-based evaluation method MedDisKEval: We prompt LLMs to retrieve information related to these clinical knowledge aspects. Then, we evaluate an LLM's mastery of medical knowledge by measuring the similarity between the LLM-generated information and the content within our knowledge base. Our experimental findings reveal that over 50% of the clinical information generated by our evaluated LLMs is significantly inconsistent with the corresponding knowledge stored in our knowledge base. We further perform a significance analysis to compare the performance of medical LLMs with their backbone models, discovering that 5 out of 6 medical LLMs perform less effectively than their backbone models in over half of the clinical knowledge aspects. These observations demonstrate that existing LLMs have not mastered adequate knowledge for clinical practice. Our findings offer novel and constructive insights for the advancement of medical LLMs.

## 1 Introduction

In recent years, advancements in Large Language Models (LLMs) have shown potential across various domains, including the medical domain. Several foundation LLMs like ChatGPT (Ouyang et al., 2022) and LLaMa (Touvron et al., 2023) have been noted for their outstanding performance on various medical evaluation benchmarks, including USMLE (United States Medical Licensing Examination) (Kung et al., 2023), the medical section of MMLU (Hendrycks et al., 2020), MedQA (Jin et al., 2021), and PubMedQA (Jin et al., 2019). However, direct application of general-purpose LLMs to the medical domain may not be suitable due to their lack of specialized training on medical corpora and potential deficits in professional expertise within the medical field. To address this gap, researchers have proposed several LLMs (Li et al., 2023; Wang et al., 2023; Chen et al., 2023; Zhang et al., 2023; Xiong et al., 2023; Singhal et al., 2023a) tailored for medical applications, known as "medical LLMs". Some of these models are claimed to outperform general LLMs like ChatGPT in specific medical tasks, such as medical dialogues and medical question answering. However, does the excellent performance achieved in these medical benchmarks and tasks indicate that current LLMs, including general and medical ones, master adequate knowledge for solving real clinical problems?

To answer this question, we need to take a throughout look at existing medical evaluation benchmarks. The existing medical evaluation benchmarks are predominantly based on question-answering (QA) tasks. These benchmarks collect questions from diverse sources, including medical examinations, electronic health records, online resources, and expert crafting. While these QA-based evaluation benchmarks are effective for assessing LLM performance, they cannot answer whether LLMs

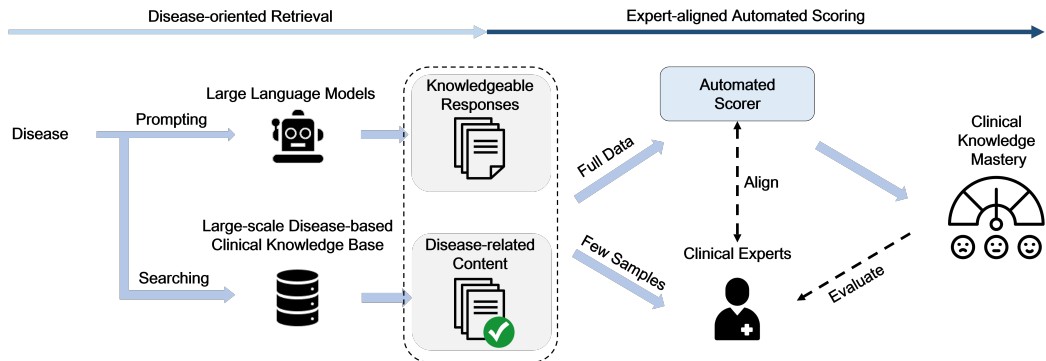

Figure 1: An overview of the proposed evaluation framework, consisting of two stages: disease-oriented clinical knowledge retrieval and expert-aligned automated scoring.

have mastered sufficient medical knowledge for solving real clinical problems. This is because current QA-based medical evaluation datasets cover only some common diseases and lack extensive coverage of knowledge across various aspects of diseases. Therefore, the performance of LLMs on these medical QA datasets cannot accurately reflect the extent to which they cover knowledge about different diseases and various knowledge aspects of diseases. Moreover, answering questions involves three distinct skills: understanding the question, mastering the relevant knowledge, and applying that knowledge for reasoning. Therefore, the performance of LLMs on QA datasets is jointly determined by these three skills and does not directly reflect their mastery of clinical knowledge. Furthermore, some of these benchmarks are available online and may be inadvertently included into the training sets of some LLMs by web crawlers or similar tools used by LLMs developers. Such data leakage may lead to unfair comparisons.

To address these shortcomings, we present in this paper a novel framework to probe whether LLMs have mastered comprehensive medical knowledge for real clinical challenges. Figure 1 presents an overview of this framework. To begin, we construct a large-scale **med**ical **dis**ease-based **k**nowledge base **MedDisK**, encompassing 10,632 common diseases and 18 clinical knowledge aspects necessary for diagnosing and treating diseases, such as primary symptoms, surgical procedures, and medications. Built on that, we propose a MedDisK-based evaluation method MedDisKEval: LLMs are first prompted to recall information of the knowledge aspects defined in our knowledge base, such as "the primary symptoms of virus URI are ..." and "the anatomy parts of diabetes are ...". The LLM's mastery of clinical knowledge is then probed by measuring the similarity between the LLM-generated disease information and the content within our knowledge base.

We perform the proposed evaluation on a total of 12 general and medical LLMs. Our experimental results indicate that, more than 50% of the disease-related information generated by all the evaluated LLMs exhibit significant inconsistencies with the content from our knowledge base (See Figure 4). **The experimental results answer our question in the first paragraph: None of the current LLMs have yet mastered adequate clinical knowledge.** Additionally, we observe that 5 out of 6 medical LLMs achieve inferior performance compared to their backbone models in over half of the clinical knowledge aspects. The results imply that the training methods applied in current medical LLMs may not consistently enhance the mastery of clinical knowledge and could potentially result in catastrophic forgetting in some knowledge aspects. To ensure the timeliness of this evaluation framework while guarding against data leaks, we will not release the complete medical knowledge base. Nevertheless, we will make data samples and an evaluation interface available at [URL to be released] to promote further research. Our contributions are summarized as follows:

- We propose a large-scale medical disease-based knowledge base MedDisK, covering 10,632 common diseases and 18 clinical knowledge aspects that are crucial for diagnosing and treating diseases.

- Built on that, we introduce a MedDisK-based evaluation method MedDisKEval to probe LLMs' mastery of clinical knowledge. Employing the proposed clinical knowledge base, we conduct an extensive evaluation of 12 LLMs to assess their clinical knowledge mastery.

- Our experimental results demonstrate that none of the evaluated LLMs have mastered sufficient knowledge to handle real clinical problems effectively. Further analysis indicates that most of the current medical LLMs do not significantly surpass their backbone models in medical knowledge mastery.

## 2 RELATED WORKS

**Medical Large Language Models** Current medical LLMs can be divided into two categories. One category supervised finetunes general backbone models with medical question answering (Singhal et al., 2023b), multi-turn medical dialogue (Zhang et al., 2023; Chen et al., 2023), data generated by LLMs (Wang et al., 2023; Li et al., 2023; Xiong et al., 2023) or a hybrid of general and medical data (OpenMEDLab, 2023). The other category, represented by PMC-LLaMA (Wu et al., 2023), conducts further pretraining on medical corpora. We primarily evaluate LLMs in the first category since only a few models are in the second category. Moreover, our evaluation is based on a Chinese clinical knowledge base, and current models in the second category present poor Chinese language capabilities in our preliminary experiments.

**Medical Evaluation Benchmarks** Existing medical LLMs are evaluated with question-answering (QA) tasks, including multi-choice QA (Jin et al., 2021; 2019) and open-ended QA (Singhal et al., 2023a; He et al., 2019). Though QA tasks are demonstrated as effective tools to evaluate LLMs' capabilities, they have limitations in measuring LLMs' medical knowledge mastery. Therefore, we propose a disease-knowledge-based evaluation that probes LLMs' proficiency in clinical knowledge. When scoring open-ended QA, automated metrics (Papineni et al., 2002; Lin, 2004; Zhang et al., 2019) are widely used but may not align well with human judgments. LLMs (OpenMEDLab, 2023) or human experts (Singhal et al., 2023b) are also employed, though incurring significant costs for comprehensive assessments. Therefore, we introduce a low-cost, expert-aligned automated scoring method to produce scores consistent with expert assessment.

**Knowledge-graph-based Language Model Evaluation** Some prior studies (Petroni et al., 2019; Sung et al., 2021) assess language models like BERT (Devlin et al., 2018) and BioBERT (Lee et al., 2020) by completing triples in knowledge graphs. While these studies probe LMs' knowledge in the general and biomedical domains, we focus on probing larger LMs in the clinical domain. We employ a large-scale clinical knowledge base including 10,632 diseases across 18 attributes to evaluate the clinical knowledge mastery of 12 LLMs.

## 3 METHODS

In this section, we present the framework to assess whether LLMs have mastered comprehensive medical knowledge for real clinical diagnosis and medical decisions, by first introducing a large-scale medical disease-based knowledge base **MedDisK** in Section 3.1 and then the MedDisK-based evaluation method **MedDisKEval** in Section 3.2.

### 3.1 MEDDISK:LARGE-SCALE MEDICAL DISEASE-BASED KNOWLEDGE BASE

As we know, disease-based clinical knowledge is of utmost importance and crucial for making accurate clinical diagnoses, conducting appropriate examinations, implementing effective treatments, and other medical decision-making. Therefore, we construct a large-scale medical disease-based knowledge base **MedDisK** to evaluate LLMs. To make the evaluation effective, the MedDisK must require the following properties:(1) including large-scale common diseases;(2) involving rich disease-based knowledge;(3) accurate and inaccessible publicly (avoiding implicit leaks leading to internal testing). To address the above issues, we employ an ICD10-based method to construct MedDisK as presented in Figure 2. ICD10 was developed by the World Health Organization (WHO), including almost all diagnosis diseases and related health problems. We first select a subset from the ICD10 database according to whether the diseases are common in clinical (determined by clinical experts) and are statistically frequent in EHR (Electronic Health Record), resulting in 10,632 common diseases. Then, we employ clinical experts to define 18 disease-based clinical knowledge aspects (in Table 1) that are crucial to medical decision-making (diagnoses, examinations, treatments). Finally, **the MedDisK, including 10,632 common diseases and their corresponding 18**

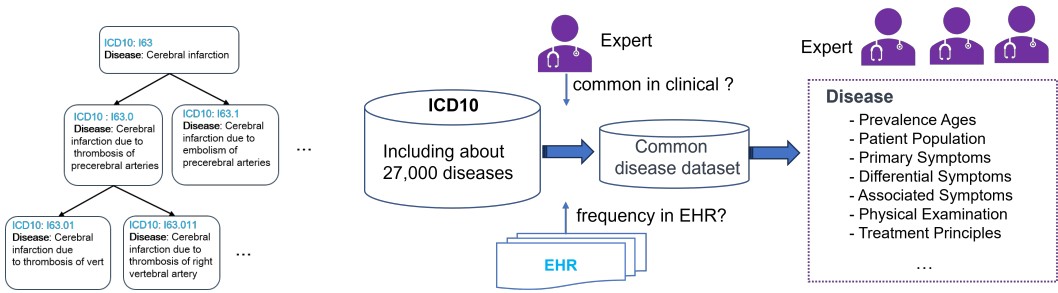

(a) An example in the ICD10.  (b) The procedure of constructing our MedDisK.

Figure 2: ICD10-based large-scale common disease clinical knowledge base construction.

| Clinical knowledge aspects | Definitions |
| --- | --- |
| Patient Population | The most group of individuals affected by the disease. |
| Prevalence Ages | The ages at which the disease commonly occurs. |
| Onset Ages | The ages at which the disease occurs exclusively. |
| Primary Symptoms | The most prominent clinical symptoms of the disease. |
| Associated Symptoms | Other clinical symptoms accompanying the primary symptoms. |
| Differential Symptoms | Specific symptoms differentiating the disease from others. |
| Physical Examinations | Physical examination results specific to the disease. |
| Anatomical Sites | Specific locations or regions on or within the body identified based on the anatomy of the disease. |
| Affected Sites | The area of the body damaged or affected by the disease. |
| Affected Body Systems | The body systems that are damaged or affected by the disease. |
| Treatment Principles | The clinical principles for developing a treatment to the disease. |
| Secondary Diseases | Possible additional diagnosis to the disease. |
| Medications | Prescribed drug(s) used to treat the disease. |
| Surgical Procedures | Medical procedures that treat the disease, involving the cutting, repairing, or removal of tissue or organs. |
| Laboratory Examinations | Abnormal laboratory examination results to the disease. |
| Auxiliary Examinations | Abnormal auxiliary examination results to this disease. |
| Departments | The specific medical departments responsible for the disease. |
| Severity Level | The severity level of the disease. |

Table 1: Definitions of clinical knowledge aspects to each disease in our MedDisK.

**aspects of clinical knowledge**, are constructed with a collaborative effort between clinical experts and machine assistance. The annotation by clinical experts ensures the accuracy, professionalism, and completeness of knowledge in MedDisK. The whole process involved the dedicated efforts of 20 clinical experts over about 10 months. More details of MedDisK construction and comparison with existing QA evaluation datassets are provided in Appendix A.

## 3.2 MEDDISKEVAL:DISEASE-KNOWLEDGE-BASED LLMS EVALUATION

### 3.2.1 DISEASE-ORIENTED CLINICAL KNOWLEDGE RETRIEVAL

We employ different prompting strategies for different categories of LLMs to extract disease-related information from each clinical knowledge aspect individually. For pretraining-only models (not fine-tuned on specific instructions), we apply the few-shot learning strategy utilized in existing benchmarks, such as MMLU, by generating prompts with five demonstrative examples. We have discovered in experiments that five examples suffice to activate the few-shot capability of LLMs. For models finetuned on instructions, we collaborate closely with clinical experts to craft tailored instructions for each knowledge aspect. These instructions are added before the few-shot examples, acknowledging that these models may achieve suboptimal performance without instructions. Each instruction is designed to introduce the relevant knowledge aspect and guide the format of LLMs'

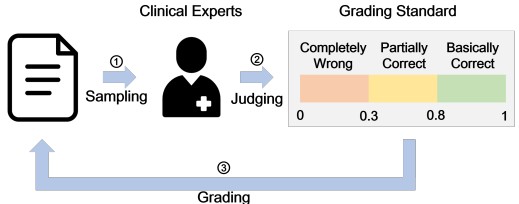

| Metrics | Correlation With Clinical Experts |
|---|---|
| BLEU-1 | 0.722* |
| ROUGE-1 | 0.779* |
| Cosine Similarity | 0.805* |
| Average | 0.837* |

Figure 3: Expert-Aligned Knowledge Scoring: Clinical experts create the grading standard based on sampled automated evaluation results.

Table 2: Consistency between the results of Med-DisKEval and clinical experts across metrics, measured by Spearman correlation coefficients. Asterisks indicate the correlations are significant.

output accordingly, undergoing multiple iterations to achieve optimal generation results. We provide prompt examples and all the instructions in Appendix B and C, respectively.

After the generation, we post-process LLM responses to remove noise and format them according to three types of clinical knowledge aspects: 1. enumerated type (a list of entities); 2. declarative type (unstructured text); 3. numeric type. We first apply heuristic rules to extract related segments and filter out irrelevant content in LLMs' responses. Afterward, we leverage various methods to format responses for different types of knowledge aspects. For the enumerated type, we employ a specialized NER model to identify and extract medical entities from the text. In cases involving the numeric type, we extract the initial number within the text and return NaN if no number is found. We do not format responses of the declarative type as they inherently assume a textual form. We denote each piece of post-processed information as a triplet $(d, a, r)$, where $d$ is the corresponding disease, $a$ is the involved clinical knowledge aspect, and $r$ is LLM's post-processed information. We provide more details of the post-processing and the NER model in Appendix D and E.

### 3.2.2 EXPERT-ALIGNED KNOWLEDGE SCORING

The proposed expert-aligned knowledge scoring process includes two steps: **disease-knowledge-based automated scoring** that assess the similarity between LLM-generated information and the content within our knowledge base using automated metrics, and **expert-aligned grading** that aligns the automated scores with expert assessment, yielding results that are more easily interpretable.

**Disease-Knowledge-based Automated Scoring** We employ automated evaluation metrics to measure the similarity between LLM-generated information and the content within our knowledge base. Firstly, for each piece of LLM-generated information $(d, a, r)$, we retrieve the corresponding triplet $(d, a, \hat{r})$ from our knowledge base. Then, the similarity is calculated as $s = sim(r, \hat{r})$, where $sim$ refers to an evaluation metric that varies according to the type of knowledge aspect $a$. For the declarative type, we apply both token-level metrics, such as BLEU-1 (Papineni et al., 2002) and ROUGE-1 (Lin, 2004) (f1-score), and a sentence-level metric cosine similarity based on a Chinese text embedding model M3E (Wang Yuxin, 2023). We have explored alternative metrics like BERTScore (Zhang et al., 2019) but found that the computed scores achieve lower consistency with expert assessment (See Appendix F). When dealing with the enumerated type, considering computational complexity, we adopt a straightforward approach by concatenating entities with blank spaces and applying the same metrics for declarative types. In the case of the numeric type, we evaluate it using the hard match score $\mathbf{1}_{r=\hat{r}}$, as our knowledge base includes only one numerical aspect (Severity Level), where distinct numbers correspond to different categories.

**Expert-aligned Grading** The disease-knowledge-based automated scoring method offers objective but less interpretable scores that reveal clinical knowledge mastery. The scores are not inherently aligned with the subjective assessments of clinical experts. Furthermore, variations in the types of knowledge aspects can introduce disparities in score distributions, thus constraining comprehensive analysis across different aspects. As a solution, we develop an expert-aligned grading approach to categorize consistency scores into distinct levels, facilitating interpretable comparisons and cross-aspect analysis. The grading process is illustrated in Figure 3. We first conduct interval sampling on all the scoring results across LLMs. Subsequently, we engage clinical experts to categorize LLM's responses into multiple tiers aligned with their subjective cognition and determine the optimal grading standard (score thresholds) that divide the results into these tiers:

- **Completely Wrong**: the LLM-generated information $r$ has a significant inconsistency or conflict with the ground truth $\hat{r}$, or even irrelevant to the aspect $a$.

- **Partially Correct**: the LLM-generated information $r$ contains some accurate information mentioned in $\hat{r}$ but may also include some incorrect or incomplete information.

- **Basically Correct**: the LLM-generated information $r$ is mostly in agreement with the ground truth $\hat{r}$. There might be minor errors or incompleteness, but the consistency is high.

Specifically, we determine a grading standard for each combination of metrics (ROUGE-1, BLEU-1, and cosine similarity) and types (enumerated and declarative). In each combination, we set a score interval of 0.1 and sample 10 examples from each interval, where each example consists of $d$, $a$, $r$, $\hat{r}$, and a similarity score $s$. For the numeric type, such as Severity Level, in our case, we directly map the score 1 to 'Basically Correct' and the score 0 to 'Completely Wrong,' as it has only two possible values. Ultimately, the clinical knowledge mastery of an LLM can be reflected by the proportion of LLM-generated information in these three tiers. More details are presented in Appendix G.

To validate the alignment between the proposed expert-aligned automated grading and expert evaluation, we assign clinical experts to annotate another 150 randomly selected instances. Each instance includes a disease $d$, a knowledge aspect $a$, information from an LLM ($r$), and $\hat{r}$ from our knowledge base. The tiers "Completely Wrong," "Partially Correct," and "Basically Correct" are mapped to respective scores of 0, 1, and 2. We employ Spearman correlation coefficients to measure the consistency and summarize results in Table 2. All three metrics achieve correlation coefficients surpassing 0.7, indicating the high consistency between the proposed automated grading and the expert assessment. Cosine similarity correlates more strongly with expert assessments than the other two metrics. However, we find that the average scores of all three metrics after grading achieve stronger correlation than any single metric (Table 2), indicating that these three metrics can complement each other in our evaluation. Therefore, we use all these metrics for a comprehensive evaluation.

## 4 EVALUATION

### 4.1 EVALUATED LLMS

As mentioned above, we evaluate two types of LLMs in our experiments: (1) LLMs that are pretrained and finetuned in general domain: GPT-3.5-turbo (Ouyang et al., 2022), Bloomz-7.1B-mt (Muennighoff et al., 2023), LLaMa-7B (Touvron et al., 2023), Vicuna-7B (Zheng et al., 2023), ChatGLM-6B (Du et al., 2022), and Baichuan-7B (Yang et al., 2023); (2) LLMs that are further finetuned on medical data: ChatDoctor (Li et al., 2023), DoctorGLM (Xiong et al., 2023), BenTsao (huatuo-llama-med-chinese) (Wang et al., 2023), HuatuoGPT (Zhang et al., 2023), BianQue-2 (Chen et al., 2023), and PULSE (OpenMEDLab, 2023). These LLMs are selected based on a comprehensive consideration of computational power, evaluation cost, and model availability. To ensure a fair comparison, we maintain the text generation parameters of LLMs as default in their respective GitHub or HuggingFace repositories.

### 4.2 RESULTS

#### 4.2.1 OVERALL PERFORMANCE

The upper part of Figure 4 depicts the distribution of all LLMs' responses across the three metrics within the three tiers defined in Section 3.2.2. These three sub-figures reveal the overall performance of current LLMs on clinical knowledge mastery. Our findings point to a striking revelation: The experimental results reveal that **over 50% of responses generated by current LLMs are classified as "Completely Wrong," approximately 30% fall under the category of "Partially Correct," and merely fewer than 20% are deemed "Basically Correct."** These results show that the clinical knowledge mastery of existing LLMs is far from adequate to address real-world clinical challenges. The distribution of the three metrics exhibits similar trends while varying in detailed proportions, highlighting the importance of utilizing multiple metrics in our evaluation. We provide some examples of LLMs' responses within three tiers in Table 3. The degree of clinical knowledge mastery shown in these LLM responses closely corresponds with the tiers assigned by our method.

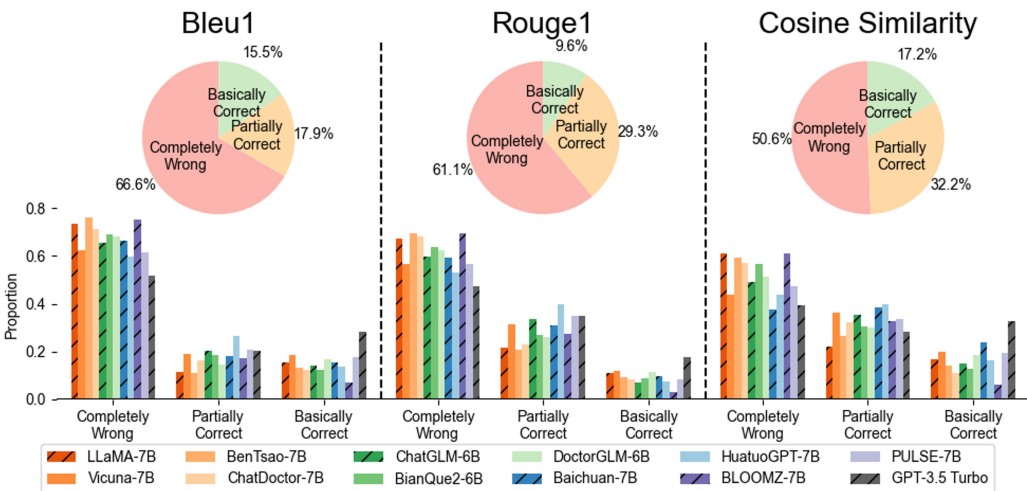

Figure 4: Upper: Distribution of LLMs' responses across three metrics. Lower: Distribution of responses across 12 LLMs and three metrics. We denote Bloomz-7.1B-mt as Bloomz-7B for name consistency with other models. Models with the same backbone model are illustrated with similar colors. We use slashes to denote the base models within each model series.

| Tier | Disease | Knowledge Aspect | Ground Truth | LLM Response |
|---|---|---|---|---|
| Completely Wrong | rheumatoid arthritis of the hand interphalangeal joints | patient population | higher prevalence in females; middle-age; elderly | ok, I see. |
| Partially Correct | tracheobronchial amyloidosis | affected sites | trachea; bronchi; lung | lung; chest |
| Basically Correct | esophageal abscess | affected body systems | digestive system | digestive system |

Table 3: Examples of LLMs' responses within three tiers defined in Section 3.2.2.

### 4.2.2 DETAILED COMPARISON ACROSS LLMS

We further investigate the clinical knowledge mastery across different LLMs by examining the distribution of different LLM's responses across three tiers, which is showcased in the lower part of Figure 4. See Appendix H for another comparison across knowledge aspects. Across all evaluated LLMs and metrics, over 40% of the clinical information generated by each LLM exhibits significant inconsistencies or conflicts with the knowledge stored in our knowledge base. This indicates that the insufficient medical knowledge mastery of existing LLMs, as demonstrated in Section 4.2.1, is not caused by a few models but is a widespread phenomenon of current LLMs. Moreover, we consider a group of LLMs using the same backbone model as an LLM series and compare the medical knowledge mastery between different series that share a similar number of parameters (excluding ChatGPT). The general order is as follows: Baichuan-7B series holds the first position, ChatGLM-6B series takes the second place, and LLaMA-7B and Bloomz-7B series share the third place.

Remarkably, GPT-3.5-turbo (ChatGPT) stands out by achieving the highest proportion of "Basically Correct" and the lowest proportion of "Completely Wrong," surpassing all other LLMs in terms of clinical knowledge mastery. Additionally, ChatGPT achieves a higher "Basically Correct" proportion than "Partially Correct" in 2 out of 3 metrics, indicating that ChatGPT exhibits lower hallucination and tends to avoid responding when faced with uncertain knowledge.

For a straightforward assessment of the clinical knowledge mastery in these LLMs, we begin by averaging the distributions of the three metrics for each LLM. Then, we perform a weighted summation across the three tiers, assigning scores of 0, 5, and 10 to "Completely Wrong," "Partially Correct," and "Basically Correct," respectively, to yield a total score for each LLM. It is worth noting that

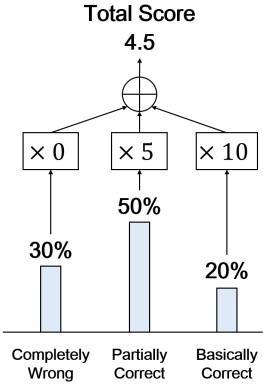

Total Score
4.5

Figure 5: Calculating total scores with the distribution on three tiers.

| Model | Type | Completely Wrong | Partially Correct | Basically Correct | Total Score | Level |
|---|---|---|---|---|---|---|
| GPT-3.5 Turbo | General | 46.1% | 27.7% | 26.1% | 4.00 | 1 |
| Vicuna-7B | General | 54.2% | 28.9% | 17.0% | 3.14 | 2 |
| Baichuan-7B | General | 54.4% | 29.3% | 16.3% | 3.10 | |
| HuatuoGPT-7B | Medical | 52.1% | 35.5% | 12.4% | 3.02 | |
| PULSE-7B | Medical | 55.2% | 29.8% | 15.1% | 3.00 | |
| DoctorGLM-6B | Medical | 60.7% | 23.6% | 15.7% | 2.75 | 3 |
| ChatGLM-6B | General | 58.0% | 29.8% | 12.1% | 2.70 | |
| BianQue2-6B | Medical | 63.1% | 25.6% | 11.4% | 2.41 | |
| LLaMA-7B | General | 67.2% | 18.3% | 14.4% | 2.36 | |
| ChatDoctor-7B | Medical | 65.5% | 23.9% | 10.6% | 2.26 | |
| BenTsao-7B | Medical | 68.3% | 19.4% | 12.2% | 2.20 | |
| BLOOMZ-7B | General | 68.7% | 25.9% | 5.5% | 1.84 | 4 |

Table 4: The ranking of evaluated LLMs based on total scores computed by the method presented in Figure 5, classified into four levels.

this score is equivalent to the average score in Table 2 that achieves high consistency with expert assessment. Subsequently, we categorize LLMs into four levels based on these total scores. The scoring process and outcomes are detailed in Figure 5 and Table 4, respectively. Surprisingly, it is evident that none of the top three models have received specialized training on medical corpora, and most medical LLMs are placed in Level 3. Additionally, models sharing the same base architecture tend to attain similar scores (e.g., LLaMA, ChatDoctor, and BenTsao; ChatGLM, DoctorGLM, and BianQue-2), although a few exceptions exist (Vicuna, PULSE). These findings suggest that most current medical LLMs perform not significantly different from their backbone models.

### 4.2.3 MEDICAL LLMS VERSUS THEIR BACKBONE MODELS

To investigate the effect of continual training on medical corpora, we further conducted a significance analysis comparing each medical LLM with its corresponding backbone model. We employed Welch's T-test to assess six model pairs across all 18 aspects of disease knowledge, utilizing the cosine similarity for analysis. The results of the T-test utilizing other metrics (ROUGE-1, BLEU-1) show similar trends and can be found in Appendix I. The findings are presented in Table 5. Within this table, the t-statistics reveal disparities in performance between medical LLMs and their backbone models across various knowledge aspects. Asterisks' presence denotes statistical significance (p-value $< 0.05$). green cells in the table signify superior performance by the medical LLM compared with its backbone model on the respective aspect, red cells indicate poorer performance, while white cells suggest no significance.

The experimental results reveal that 5 out of 6 medical LLMs underperform significantly compared to their base models in over half of the clinical knowledge aspects. PULSE stands out as the sole model achieving significant improvements on almost all evaluated aspects except the Severity Level. The significant improvement attained by the PULSE model can be attributed to its finetuning on approximately 4,000,000 instructions from both the Chinese medical field and the general domain. However, this significant improvement may also be affected by the low performance of its backbone model, Bloomz-7.1B-mt, on the proposed evaluation benchmark (see Figure 4). Medical LLMs typically excel in certain aspects, such as Patient Population and Departments, but exhibit subpar performance in other areas, such as Anatomical Sites and Secondary Diseases.

In summary, the results imply that most of the current medical LLMs do not achieve consistent enhancement in the clinical knowledge mastery across all knowledge aspects compared to their backbone models, even potentially resulting in catastrophic forgetting in some aspects.

## 5 DISCUSSION

**Medical Capabilities of Current LLMs** Large Language Models cannot be widely employed in real clinical tasks unless they master adequate clinical knowledge, exceptional medical compre-

| Backbone Models | ChatGLM-6B | | Bloomz-7B | Baichuan-7B | LLaMA-7B | |
|---|---|---|---|---|---|---|
| Medical LLMs | BianQue-2 | DoctorGLM | PULSE | HuatuoGpt | BenTsao | ChatDoctor |
| Patient Population | 15.0* | 36.7* | 27.0* | 24.2* | 40.2* | 31.6* |
| Prevalence Ages | 29.1* | 63.7* | 3.7* | -23.0* | -35.0* | -9.4* |
| Onset Ages | 68.0* | 173.1* | 17.9* | -112.2* | -103.0* | -0.6 |
| Primary Symptoms | -23.8* | -38.8* | 87.4* | 14.9* | -3.2* | -10.3* |
| Associated Symptoms | 2.8* | -7.4* | 13.6* | -2.1* | -5.4* | 9.9* |
| Differential Symptoms | -38.0* | -26.0* | 28.0* | 9.8* | -7.0* | 7.2* |
| Physical Examination | -53.8* | -14.5* | 13.8* | -29.6* | -7.4* | -27.4* |
| Anatomical Sites | -55.0* | -16.0* | 83.9* | -41.2* | -15.3* | -92.0* |
| Affected Sites | -41.8* | -11.6* | 56.0* | -29.3* | -32.5* | -41.4* |
| Affected Body Systems | -76.9* | -62.7* | 38.1* | 20.9* | 58.0* | 81.1* |
| Treatment Principles | 2.6* | 11.7* | 15.8* | 2.2* | 15.2* | -9.6* |
| Secondary Diseases | -13.1* | -45.8* | 22.9* | -28.4* | -42.8* | -27.0* |
| Medications | 11.9* | 4.9* | 21.1* | -17.7* | 41.2* | 25.7* |
| Surgical Procedures | -8.3* | -4.7* | 12.8* | -5.4* | -4.8* | -6.2* |
| Auxiliary Examinations | -35.1* | -9.7* | 9.4* | -27.7* | -10.6* | -28.3* |
| Laboratory Examinations | -30.6* | -12.9* | 25.1* | -10.4* | -4.5* | -11.6* |
| Departments | 4.4* | 5.0* | 106.6* | -12.4* | 56.8* | 72.1* |
| Severity Level | 17.9* | -18.5* | -41.3* | 43.3* | 86.7* | 4.5* |

Table 5: The results of Welch's T-test between each medical LLM and its backbone model across different aspects of diseases. The cosine similarities are applied in this analysis.

hension, and strong reasoning capabilities. Among these capabilities, sufficient clinical knowledge forms the foundation for the other two. Nevertheless, our experimental results demonstrate that all current LLMs are far from mastering adequate clinical knowledge.

**Performance of Current Medical LLMs** Though several medical LLMs are claimed to perform better than their backbone models on medical evaluation benchmarks, our evaluation results indicate that they do not achieve consistent improvement in all clinical knowledge aspects, even degrading severely in some aspects. Moreover, these medical LLMs achieve inferior performance than some general LLMs with a similar number of parameters, such as Baichuan-7B and Vicuna-7B. Several factors may contribute to this phenomenon: 1. These medical LLMs have not undergone extensive pretraining on medical corpora; 2. Certain medical LLMs are trained for limited medical tasks and lack comprehensive training on diverse medical tasks; 3. The performance of a few medical LLMs may be inflated due to potential data leakage.

**Future Works** Medical LLMs have to master sufficient clinical knowledge first to become a foundation model in the medical domain. Our experiments on current medical LLMs indicate that small-scale finetuning on a limited set of medical tasks cannot inject adequate clinical knowledge into LLMs. Large-scale pretraining on medical corpora and supervised finetuning across various medical tasks may offer promising ways for training foundational models in the medical domain.

## 6 CONCLUSION

We present in this paper an evaluation framework to assess the clinical knowledge mastery of LLMs. Firstly, we construct a large-scale Chinese medical disease-based knowledge base MedDisK, covering 10,632 common diseases and 18 clinical knowledge aspects that are essential in clinical practice. Built on that, we introduce a MedDisK-based evaluation method MedDisKEval, utilizing the proposed clinical knowledge base to study the medical knowledge mastery of 12 general and medical LLMs. Our experimental results reveal that current LLMs have not mastered adequate clinical knowledge, indicating that they are not well prepared to serve as foundation models in the medical domain. A further in-depth study reveals that most current medical LLMs have not performed significantly better than their backbone models. In the future, we will continue maintaining the knowledge base we have introduced to ensure its accuracy and professionalism and support more languages to facilitate the research of this field.

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

# A    DETAILS OF MEDDISK CONSTRUCTION

## A.1    SUPPLEMENTARY OF MEDDISK CONSTRUCTION

The construction of MedDisK involves a total of two phases: selection of diseases, and knowledge annotation. In phase 1, we first conduct a statistical analysis on the occurrence of 27,000 ICD-10 diseases in 4 million highly de-identified electronic health records (EHRs) from over 100 hospitals across 5 cities. Then we selected diseases with a frequency $> 10^{-4}$, resulting in 1,048 diseases. To broaden the coverage of MedDisK, we further requested clinical experts to choose a subset of clinically important diseases from the remaining, resulting in another 9,584 diseases.

In phase 2, we employed a retrieve-and-proofread knowledge annotation method. We first exploit an information retrieval module that retrieves disease-related information from medical books and literature. Subsequently, we requested clinical experts to proofread the retrieved information and supplement missing knowledge. We find that such human-machine collaboration is helpful for minimizing human bias introduced in annotation: we have requested two experts to annotate the knowledge related to 20 diseases (involving around 1,000 disease-related knowledge points) using the human-machine collaborative method introduced above, and the results revealed a disagreement rate of less than 2%, indicating the reliability and effectiveness of our knowledge base construction method. The statistics of MedDisK, including the frequency of disseases in EHRs and the number of unique entities, are presented in Figure 6 and Table 6, respectively.

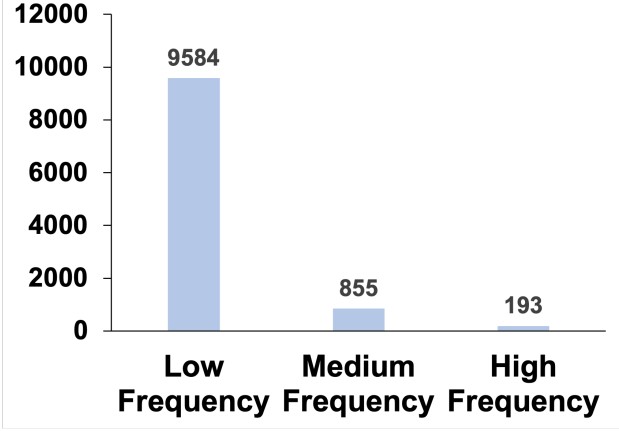

| Knowledge Aspects | Total Amount |
|---|---|
| Patient Population | 701 |
| Prevalence Ages | 146 |
| Onset Ages | 17 |
| Primary Symptoms | 16884 |
| Associated Symptoms | 4619 |
| Differential Symptoms | 12749 |
| Anatomical Sites | 1345 |
| Affected Sites | 1021 |
| Treatment Principles | 3526 |
| Secondary Diseases | 638 |
| Surgical Procedures | 5097 |
| Medications | 3826 |
| Departments | 89 |
| Affected Body Systems | 89 |

Figure 6: Frequency of diseases covered by MedDisk in 4 million EHRs. Low: frequency $< 10^{-4}$, Medium: $10^{-4} \leq$ frequency$< 10^{-3}$, High: frequency $\geq 10^{-3}$.

Table 6: Number of unique entities across 14 enumerated-type knowledge aspects.

## A.2 MEDDISK VERSUS OTHER MEDICAL DATASETS

We have conducted a comprehensive comparison between MedDisK and existing medical QA datasets in terms of coverage of common diseases, disease-based knowledge and public availability. For QA datasets, we leverage Medical Concept Annotation Tool (Kraljevic et al., 2021) to identify all the diseases and count the number of diseases in each QA dataset. We included in the comparison all QA datasets in MultiMedQA (Singhal et al., 2023b) except PubMedQA, because PubMedQA is primarily a reading comprehension dataset where answers can be directly derived from the provided context. Consequently, it is not designed to evaluate models' mastery of medical knowledge.

The results listed in Table 7 demonstrate that our proposed database covers a significantly larger amount of diseases and more types of disease-based knowledge than existing QA-based datasets. It is also worth noting that most of existing medical databases have released labeled data, which may result in data contamination that some LLMs have seen the test set of these datasets in the training phase. In contrast, we will release evaluation interface instead of the whole database to balance both the public accessibility of our evaluation and the reduction of data contamination.

| Datasets | Type | # diseases | Publicly available? |
|---|---|---|---|
| MedQA | QA dataset | 1391 | Yes |
| MedMCQA | QA dataset | 3475 | Yes |
| MMLU (medical) | QA dataset | 383 | Yes |
| MedicationQA | QA dataset | 172 | Yes |
| LiveQA | QA dataset | 480 | Yes |
| HealthSearchQA | QA dataset | 262 | Yes |
| **Total of Above** | QA datasets | 3907 | Yes |
| **MedDisK (Ours)** | Knowledge base | **10632** | Yes (evaluation platform) |

Table 7: Comparison of existing medical evaluation datasets across the number of diseases and public availability.

Results in Table 8 show that existing medical QA evaluation sets have not covered as many disease-knowledge-related entities as MedDisK does. It is worth noting that MedDisK covers even more entities than the sum of these 6 medical QA datasets, indicating that our evaluation benchmark obtains a much broader coverage of disease-related clinical knowledge than existing medical evaluation benchmarks.

| Dataset | #Popu. | #Symp. | #Part. | #Syst. | #Proc. | #Medi. | #Dept. |
|---|---|---|---|---|---|---|---|
| MedQA | 197 | 377 | 574 | 15 | 429 | 62 | 36 |
| MedMCQA | 241 | 452 | 1245 | 33 | 811 | 56 | 54 |
| MMLU (medical) | 92 | 114 | 250 | 10 | 111 | 6 | 9 |
| MedicationQA | 27 | 64 | 52 | 7 | 70 | 67 | 3 |
| LiveQA | 75 | 108 | 141 | 9 | 166 | 14 | 19 |
| HealthSearchQA | 10 | 63 | 40 | 3 | 4 | 2 | 2 |
| **Total of Above** | 349 | 570 | 1362 | 34 | 997 | 183 | 83 |
| **MedDisK (Ours)** | **701** | **18737** | **1585** | **89** | **5097** | **3826** | **89** |

Table 8: Comparison of existing medical QA datasets with the proposed MedDisK database across 7 medical entities, including patient population (Popu.), symptoms (Symp.), body parts (Part.), body systems (Syst.), therapeutic procedure (Proc.), medication (Medi.), and departments (Dept.). Note that for MedDisK, we count unique anatomic/affected sites for the number of body parts, and unique primary/associated/differential symptoms for the number of symtoms.

# B    DETAILS OF LLMS' KNOWLEDGE RETRIEVAL

We employ distinct prompting strategies tailored to various types of LLMs. For pretraining models, we provide a set of five examples preceding the question. In the case of instruction-tuning models, our approach begins with explicit instructions detailing the knowledge aspect, followed by five illustrative examples, and culminates with the question itself. Exemplars of these prompts are showcased in Figure 7, while a comprehensive compilation of instructions for each knowledge aspect is available in Appendix C.

**Pretraining Models:**

先天性核性白内障的常见症状包括白瞳症；视力低下;斜视;…。甲状腺激素抵抗综合征的常见症状包括甲状腺肿大;水肿;乏力;心动过缓;…。上呼吸道感染的常见症状包括咳嗽;打喷嚏;流鼻涕;...

The primary symptoms of congenital nuclear cataracts include leukocoria; reduced vision; strabismus; ... The primary symptoms of thyroid hormone resistance syndrome include thyroid enlargement; swelling; fatigue;... The primary symptoms of upper respiratory tract infections include coughing; sneezing; runny nose; ...

**Instruction Tuning Models:**

请输出急性前壁心肌梗死在辅助检查中表现的异常检查结果。辅助检查涉及各类除实验室化验类的检查，如：医技、测量表等。请按照"检查项目＋（检查阳性结果）"的格式书写，注意使用括号括住检查结果，可能存在多个检查项目及结果。若不存在请只输出"无"。例如：先天性核性白内障的辅助检查异常结果为晶状体裂隙灯检查（晶状体浑浊）。甲状腺激素抵抗综合征的辅助检查异常结果为……
请回答：急性前壁心肌梗死的辅助检查异常结果为?
答案：急性前壁心肌梗死的辅助检查异常结果为：心电图（ST段抬高、T波倒置、Q波出现）……

Please provide the abnormal results for acute anterior wall myocardial infarction in auxiliary examinations. Auxiliary examinations encompass various types of tests apart from laboratory assays, such as medical technology, measurements, etc. Please format the responses as "Test Name (+ Positive Result)" and use parentheses to enclose the test results. Multiple test items and results may exist. If there are no abnormal findings, please only state "None." For example: The abnormal result in auxiliary examinations for congenital nuclear cataracts is slit-lamp examination (opacification of the lens). The abnormal result in auxiliary examinations for thyroid hormone resistance syndrome is … Please answer: What are the abnormal results in auxiliary examinations for acute anterior wall myocardial infarction?
**Answer:** The abnormal results in auxiliary examinations for acute anterior wall myocardial infarction is electrocardiogram (ST-segment elevation, T-wave inversion, Q-wave appearance)...

Figure 7: Examples of the disease-oriented clinical knowledge retrieval process in the proposed evaluation method, each includes a Chinese prompt (used in our experiments) and its English translation. The few-shot examples are directly extracted from our knowledge base and transformed into text with templates. The blue text is the test sample, and the violet underlined text is the response from LLMs.

## C LIST OF INSTRUCTIONS

| Knowledge Aspect | Prompt |
|---|---|
| 人群
Patient Population | 请尽可能多地列举{}的常见患病人群。常见患病人群可能涉及某疾病的高发年龄段、患有某病史人群、特定性别人群。请使用分号;分隔这些人群，若无明确常见患病人群请只输出"无"
Please list as many common patient populations for {}. Common patient populations may encompass age groups with a high incidence of a particular disease, individuals with a history of a specific illness, and gender-specific groups. Please separate these populations with semicolons (;). If there are no clearly defined common patient populations, please only output "None." |
| 好发年龄
Prevalence Ages | 请尽可能多地列举{}的好发年龄。可能的年龄段有围生期、新生儿、婴儿、幼儿、儿童、少年、成人、青年、中年、老年。请使用分号;分隔这些年龄段，若未明确哪个年龄段请只输出"无"
Please provide as many prevalence ages for {} as possible. Possible age groups include perinatal, neonatal, infant, toddler, child, adolescent, adult, young adult, middle-aged, and elderly. Please separate these age groups with semicolons (;). If it is not clear which age group is applicable, please only output "None." |
| 特发年龄
Onset Ages | 请尽可能多地列举{}的特发年龄。特发年龄指的是该疾病只在某一年龄段发作，在其他年龄段不发作。可能的年龄段有围生期、新生儿、婴儿、幼儿、儿童、少年、成人、青年、中年、老年。请使用分号;分隔这些年龄段，若本病未明确哪个特发年龄段请只输出"无"
Please provide as many onset ages for {} as possible. Onset ages refer to instances where a disease occurs exclusively in a certain age range and not in others. Possible age groups include perinatal, neonatal, infant, toddler, child, adolescent, adult, young adult, middle-aged, and elderly. Please separate these age groups with semicolons (;). If it is not clear which specific age group is applicable for the given condition, please only output "None." |
| 症状
Primary Symptoms | 请尽可能多地列举{}的常见症状。症状是疾病引起病人主观上的异常感觉或某些客观病态改变。请使用分号;分隔这些特发症状。
Please list as many primary symptoms of {} as possible. Symptoms refer to subjective abnormalities experienced by patients or certain objective pathological changes caused by the disease. Please separate these specific symptoms with semicolons (;). |
| 伴随症状
Associated Symptoms | 请尽可能多地列举{}的伴随症状。伴随症状是因为某疾病引起一系列主要症状时，继而出现的一些别的不适反应；是疾病次要影响某器官产生的症状。请使用分号;分隔这些特发症状，若本病没有明确的伴随症状请只输出"无"
Please provide as many associated symptoms of {} as possible. Associated symptoms are additional discomforts or secondary manifestations that occur as a result of a disease causing a series of primary symptoms or affecting a specific organ. Please separate these specific associated symptoms with semicolons (;). If the condition does not have clearly defined associated symptoms, please only output "None." |
| 鉴别性症状
Differential Symptoms | 请尽可能多地列举{}的鉴别性症状。鉴别性症状是本疾病主要症状，代表本疾病的特殊症状，疾病主要影响某器官产生的症状。请使用分号;分隔这些特发症状，若本病没有明确的伴随症状请只输出"无"
Please list as many differential symptoms of {} as possible. Differential symptoms are the primary symptoms of the disease, representing the specific features of the condition, and are symptoms primarily associated with the affected organ. Please separate these differential symptoms with semicolons (;). If the condition does not have clearly defined distinctive symptoms, please only output "None." |
| 体格检查
Physical Examination | 请输出{}的体格检查。体格检查是医生给病人检查时对本疾病具有诊断意义的体征。如果为肺部疾病，肺部体格检查书写格式如下：肺听诊（）肺视诊（）肺叩诊（）肺触诊（）；如果为腹部疾病，腹部体格检查书写格式如下：腹部触诊（）腹部听诊（）；如果为泌尿外科疾病，体格检查可写：肛门视诊（）。
Please provide the results of physical examination of {}. A physical examination is a medical assessment performed by a physician to identify clinically significant signs related to the specific disease. If it is a pulmonary condition, the format for recording a pulmonary physical examination is as follows: Pulmonary auscultation (), Pulmonary inspection (), Pulmonary percussion (), Pulmonary palpation (); If it is an abdominal condition, the format for recording an abdominal physical examination is as follows: Abdominal palpation (), Abdominal auscultation (); If it is a urological condition, the physical examination may include: Anoscopic inspection (). |
| 解剖部位
Anatomical Sites | 请尽可能多地列举{}的解剖部位。解剖部位是疾病最直接影响的部位。请使用分号;分隔这些部位。
Please list as many anatomical sites affected by {} as possible. Anatomical sites refer to the areas most directly impacted by the disease. Please separate these sites with semicolons (;). |
| 影响部位
Affected Sites | 请尽可能多地列举{}的影响部位。影响部位是疾病影响机体的所有部位。请使用分号;分隔这些部位。
Please list as many affected sites of {} as possible. Affected sites refer to all the areas of the body that are impacted by the disease. Please separate these sites with semicolons (;). |

Table 9: Instructions employed for each knowledge aspect, comprising the Chinese version and English translation. {} in instructions will be filled with disease name during experiment.

We meticulously crafted instructions for each knowledge aspect, encompassing both a detailed description of the knowledge element and specific constraints regarding the output format. Importantly,

| Knowledge Aspect | Prompt |
|---|---|
| 人体系统
Affected Body Systems | 请尽可能多地列举{}累及的相关人体系统，请使用分号;分隔这些人体系统。
Please provide as many human body systems affected by {} as possible. Please separate these body systems with semicolons (;). |
| 治疗原则
Treatment Principles | 请尽可能多地列举{}的治疗原则。输出治疗原则时尽量细化，输出具有鉴别意义的治疗作用，例如：护肝、护胃、止泻、止痛等。请使用分号;分隔这些治疗原则。
Please list as many treatment principles for {} as possible. When providing treatment principles, please be as specific as possible, mentioning distinct therapeutic actions that are diagnostically significant, such as liver protection, gastric protection, diarrhea control, pain relief, and so on. Please separate these therapeutic principles with semicolons (;). |
| 继发疾病
Secondary Diseases | 请尽可能多地列举{}的常见继发疾病。继发疾病是出现的与原发疾病有直接因果关系的疾病或病征的统称，这些疾病的转归也直接与原发病的变化有关。请注意：继发症与后遗症、并发症并不相同，不能混淆。请使用分号;分隔这些疾病，若无明确继发疾病请只输出"无"
Please list as many common secondary diseases of {} as possible. Secondary diseases refer to a collective term for diseases or signs that have a direct causal relationship with the primary disease, and their course is directly related to changes in the primary condition. Please note that secondary diseases are distinct from sequelae and complications and should not be confused. Please separate these diseases with semicolons (;). If there are no clearly defined secondary diseases, please only output "None." |
| 药物
Medications | 请尽可能多地列举治疗{}的常见药物名称。请列出具体的药物名称。请使用分号;分隔这些药物,没有请只输出"无"
Please list as many common medications used to treat {} as possible. Please provide specific drug names, and separate them with semicolons (;). If there are none, please only output "None." |
| 手术
Surgical Procedures | 请尽可能多地列举{}的常见手术治疗名称。请使用分号;分隔这些手术，没有请只输出"无"
Please list as many common surgical procedure names for {} as possible. Please separate these surgical procedures with semicolons (;). If there are none, please only output "None." |
| 辅助检查异常结果
Auxiliary Examinations | 请输出{}在辅助检查中表现的异常检查结果。辅助检查涉及各类除实验室化验类的检查，如：医技、测量表等。请按照"检查项目＋（检查阳性结果）"的格式书写，注意使用括号括住检查结果，可能存在多个检查项目及结果。若不存在请只输出"无"。
Please provide the abnormal findings of {} in auxiliary examinations. Auxiliary examinations encompass various types of tests and measurements, excluding laboratory assays. Please format the information as "test item (+ positive result)" and enclose the test result in parentheses. There may be multiple test items and results. If there are no abnormal findings, please only output "None." |
| 实验室检查异常结果
Laboratory Examinations | 请输出{}在实验室检查中的异常检查结果。请按照"检查项目＋（检查阳性结果）"的格式书写，注意使用括号括住检查结果，可能存在多个检查项目及结果。若不存在请只输出"无"。
Please provide the abnormal results of laboratory examinations for {}. Please format the information as "test item (+ positive result)" and enclose the test result in parentheses. There may be multiple test items and results. If there are no abnormal findings, please only output "None." |
| 科室
Departments | 请尽可能多地列举{}涉及的科室，请使用分号;分隔这些科室，没有请只输出"无"
Please list as many departments involved in {} as possible. Please separate these departments with semicolons (;). If there are none, please only output "None." |
| 危重等级
Severity Level | 请按照如下分级标准，评估{}的危重等级属于哪一级？危重等级分为：1级：正在或即将发生的生命威胁或病情恶化，需要立即进行积极干预；2级：病情危重或迅速恶化，如短时间内不能进行治疗则危及生命或造成严重的器官功能衰竭；或者短时间内进行治疗可对预后产生重大影响，比如溶栓、解毒等；3级：存在潜在的生命威胁，如短时间内不进行干预，病情可进展至威胁生命或产生十分不利的结局；4级：存在潜在的严重性，慢性或非常轻微的症状。输出对应的等级序号即可。
Please assess the severity level of {} according to the following grading criteria. Critical severity levels are divided as follows: Level 1: Immediate or imminent life-threatening situations or worsening conditions that require immediate and aggressive intervention. Level 2: Critical condition or rapidly deteriorating status where a delay in treatment could be life-threatening or lead to severe organ dysfunction within a short time frame. Rapid treatment within a short period can have a significant impact on prognosis, such as thrombolysis, antidote administration, etc. Level 3: Potential life-threatening situations where without intervention within a short timeframe, the condition may progress to a life-threatening state or result in highly unfavorable outcomes. Level 4: Existence of potential seriousness, chronicity, or very mild symptoms. Please provide the corresponding severity level number. |

Table 10: Instructions employed for each knowledge aspect, comprising the Chinese version and English translation. {} in instructions will be filled with disease name during experiment.

these prompts were developed in collaboration with domain experts in the medical field to ensure their high quality.

# D   DETAILS OF POST-PROCESSING

<disease name>的<knowledge aspect>为/包括___
<knowledge aspect> of <disease name> is/include ___

Figure 8: The pattern extracted from LLM's responses using regular expression, wherein <disease name >and <knowledge aspect >correspond to the question.

We leverage simple heuristic rules to extract relevant segments in the original responses, as illustrated in Figure 8. Given the disease name and the knowlegde aspect, we search patterns that appear in our crafted instructions and demonstrative examples and extract the answer part in patterns. The response will remain unchanged if no pattern is discovered in the response.

# E   DETAILS OF THE NER TOOL

The NER model that we leverage to process responses of enumerated types is trained and constructed following the method proposed in Su et al. (2022). We first pretrained a BERT-base model on 3.5 million highly de-identified EHRs from 7 hospitals with MLM objective proposed in Devlin et al. (2018). Then we finetuned the model on 200k labeled EHR segments by following the method proposed in Su et al. (2022), teaching the model to extract medical entities in EHRs. On a test set of 10k+ real-world EHRs involving 40k+ medical entities, our NER model achieves 0.88 micro-f1 score across a total of 116 types of medical entities, even surpassing 0.9 on several common medical entities, such as anatomical sites, symptoms, medication. This model has already been applied in a wide range of medical scenarios, including assisted consultations and diagnosis, as well as EHR-based semantic parsing, demonstrating consistent and reliable performance.

# F   CONSISTENCY BETWEEN AUTOMATED SCORE AND CLINICAL EXPERT

| Metrics | Correlation with Clinical Experts |
|---------|-----------------------------------|
| BLEU-1 | 0.757* |
| ROUGE-1 | 0.772* |
| Cosine Similarity | 0.762* |
| BERTScore | 0.650* |

Table 11: Consistency between clinical experts and automated score prior to expert alignment, measured by Spearman correlation. Asterisks denote statistically significance (p-value < 0.05). The correlation coefficient of BERTScore is below 0.7, indicating misalignment with human judgment.

# G   EXPERT-ALIGNED GRADING THRESHOLD

| Metrics | Type | Completely Wrong | Partially Correct | Basically Correct |
|---------|------|------------------|-------------------|-------------------|
| BLEU-1 | enumerate | [0, 0.05) | [0.05, 0.25) | [0.25, 1] |
| | declarative | [0, 0.05) | [0.05, 0.45) | [0.45, 1] |
| ROUGE-1 | enumerate | [0, 0.05) | [0.05, 0.75) | [0.75, 1] |
| | declarative | [0, 0.05) | [0.05, 0.55) | [0.55, 1] |
| Cosine Similarity | enumerate | [-1, 0.35) | [0.35, 0.75) | [0.75, 1] |
| | declarative | [-1, 0.55) | [0.55, 0.65) | [0.65, 1] |

Table 12: Thresholds used in expert-aligned automated grading. Distinct thresholds are applied to various metrics and types. Note that the grading of numetric types is relied on exact matching.

## H  PERFORMANCE ACROSS KNOWLEDGE ASPECTS

We further analysis the detailed performance of LLMs across 18 knowledge aspects, and present the results in Figure 9 and 10, respectively. The results in Figure 9 indicates that LLMs achieve relatively better performance on Treatment Principles and Onset Ages, and poorer performance on several aspects such as Secondary Diseases and Laboratory Examination.

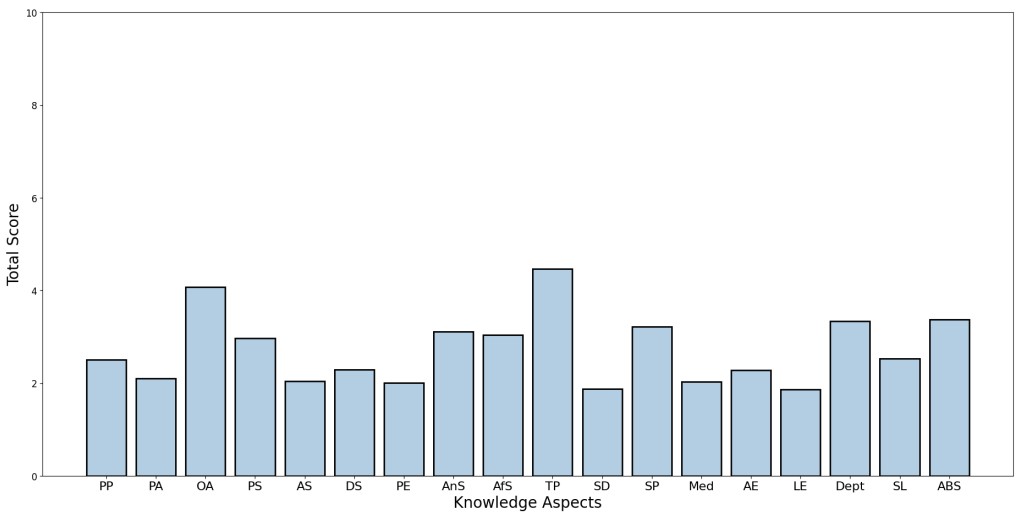

Figure 9: The average performance of LLMs across 18 clinical knowledge aspects. PP: Patient Population; PA: Prevalence Ages; OA: Onset Ages; PS: Primary Symptoms; AS: Associated Symptoms; DS: Differential Symptoms; PE: Physical Examination; AnS: Anatomical Sites; AfS: Affected Body Systems; TP: Treatment Principles; SD: Secondary Diseases; SP: Surgical Procedures; Med: Medications; AE: Auxiliary Examinations; LE: Laboratory Examinations; Dept: Departments; SL: Severity Level; ABS: Affected Body System.

Results in Figure 10 further suggest that different LLMs perform distinctly on the same knowledge aspect. For example, GPT-3.5-turbo achieves around 5 on Primary Symptoms (PS), while models such as LLaMA, BenTsao, and ChatDoctor achieve under 1 on this aspect. It is worth noting that GPT-3.5-turbo achieves relatively stable performance across all knowledge aspects.

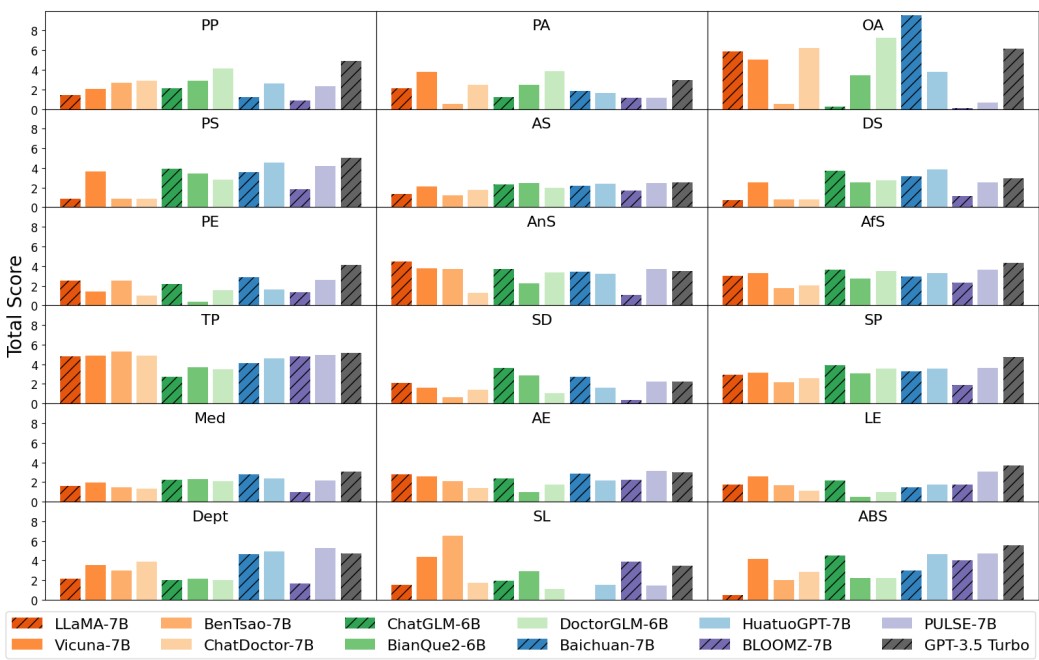

Figure 10: Performance of LLMs across 18 clinical knowledge aspects. PP: Patient Population; PA: Prevalence Ages; OA: Onset Ages; PS: Primary Symptoms; AS: Associated Symptoms; DS: Differential Symptoms; PE: Physical Examination; AnS: Anatomical Sites; AfS: Affected Body Systems; TP: Treatment Principles; SD: Secondary Diseases; SP: Surgical Procedures; Med: Medications; AE: Auxiliary Examinations; LE: Laboratory Examinations; Dept: Departments; SL: Severity Level; ABS: Affected Body System.

# I  SIGNIFICANCE ANALYSIS

| Backbone Models | ChatGLM-6B | | Bloomz-7B | Baichuan-7B | LLaMA-7B | |
|---|---|---|---|---|---|---|
| Medical LLMs | BianQue-2 | DoctorGLM | PULSE | HuatuoGpt | BenTsao | ChatDoctor |
| Patient Population | 16.2* | 39.7* | 36.8* | 40.3* | 43.9* | 43.5* |
| Prevalence Ages | 36.3* | 64.1* | 9.0* | -7.4* | -37.8* | -0.8 |
| Onset Ages | 66.6* | 155.4* | 25.3* | -111.8* | -102.9* | 5.4* |
| Primary Symptoms | -5.6* | -23.9* | 43.9* | 45.9* | -0.4 | 7.9* |
| Associated Symptoms | -0.1 | -3.6* | 24.7* | 13.4* | 4.2* | -6.5* |
| Differential Symptoms | -11.3* | -14.0* | 23.3* | 26.9* | 7.4* | -4.7* |
| Physical Examination | -44.1* | -16.7* | 23.9* | -29.2* | 10.8* | -52.5* |
| Anatomical Sites | -20.9* | -3.0* | 36.6* | -24.3* | -10.6* | -68.5* |
| Affected Sites | -27.2* | -3.1* | 22.6* | -16.1* | -19.2* | -29.9* |
| Affected Body Systems | -57.7* | -41.4* | 31.7* | 6.6* | 32.1* | 39.8* |
| Treatment Principles | 24.6* | 21.5* | -1.4 | 12.5* | 19.7* | -3.3* |
| Secondary Diseases | -12.0* | -48.9* | 40.2* | -21.8* | -35.0* | -14.2* |
| Medications | -12.5* | -14.6* | 36.4* | 10.4* | -35.2* | -26.5* |
| Surgical Procedures | -20.6* | -3.7* | 61.5* | 19.7* | -30.0* | -7.3* |
| Auxiliary Examinations | -29.0* | -17.4* | 18.2* | 9.1* | -7.2* | -38.7* |
| Laboratory Examinations | -49.1* | -46.0* | 37.5* | 25.9* | 1.5 | -25.0* |
| Departments | -11.0* | -4.0* | 74.2* | -6.2* | 0.8 | 11.5* |
| Severity Level | 17.9* | -18.5* | -41.3* | 43.3* | 86.7* | 4.5* |

Table 13: The results of Welch's T-test between each medical LLM and its backbone model across different aspects of diseases. The BLEU-1 similarities are applied in this analysis. Values in the table are the t-statistics that measure the difference of average scores between the medical LLM and its backbone model on different aspects. Asterisks indicate that the differences are significant (p-value<0.05). Note that we denote Bloomz-7.1B-mt as Bloomz-7B for name consistence with other backbone models.

| Backbone Models | ChatGLM-6B | | Bloomz-7B | Baichuan-7B | LLaMA-7B | |
|---|---|---|---|---|---|---|
| Medical LLMs | BianQue-2 | DoctorGLM | PULSE | HuatuoGpt | BenTsao | ChatDoctor |
| Patient Population | 15.6* | 39.1* | 35.4* | 39.8* | 40.4* | 40.9* |
| Prevalence Ages | 32.7* | 58.8* | 3.6* | -4.1* | -36.3* | 1.3 |
| Onset Ages | 66.5* | 155.0* | 24.8* | -111.6* | -102.9* | 5.4* |
| Primary Symptoms | -17.5* | -25.5* | 61.6* | 38.2* | 2.6* | -2.2* |
| Associated Symptoms | 0.5 | -4.4* | 24.7* | 13.5* | 5.2* | -5.7* |
| Differential Symptoms | -15.4* | -13.4* | 27.1* | 26.7* | 5.1* | -8.4* |
| Physical Examination | -59.5* | -29.4* | 29.0* | -17.9* | 8.5* | -68.7* |
| Anatomical Sites | -26.3* | -3.4* | 48.4* | -18.9* | -10.7* | -70.1* |
| Affected Sites | -28.9* | -2.7* | 25.3* | -16.6* | -21.5* | -33.5* |
| Affected Body Systems | -56.3* | -36.6* | 33.3* | 0.3 | 34.4* | 43.1* |
| Treatment Principles | 27.6* | 30.4* | -16.7* | 10.5* | 24.8* | -18.6* |
| Secondary Diseases | -12.1* | -48.9* | 39.9* | -21.9* | -34.9* | -14.2* |
| Medications | -11.7* | -13.1* | 38.0* | 6.3* | -34.5* | -26.0* |
| Surgical Procedures | -21.3* | -4.2* | 62.1* | 20.4* | -28.8* | -8.5* |
| Auxiliary Examinations | -38.2* | -25.4* | 9.3* | 11.3* | 6.0* | -50.9* |
| Laboratory Examinations | -56.2* | -52.0* | 27.6* | 28.7* | 6.3* | -28.4* |
| Departments | -11.1* | -0.2 | 84.5* | -12.6* | 10.0* | 10.4* |
| Severity Level | 17.9* | -18.5* | -41.3* | 43.3* | 86.7* | 4.5* |

Table 14: The results of Welch's T-test between each medical LLM and its backbone model across different aspects of diseases. The ROUGE-1 similarities are applied in this analysis. Values in the table are the t-statistics that measure the difference of average scores between the medical LLM and its backbone model on different aspects. Asterisks indicate that the differences are significant (p-value<0.05). Note that we denote Bloomz-7.1B-mt as Bloomz-7B for name consistence with other backbone models.

## J AN EXAMPLE OF KNOWLEDGE ASPECTS

| Clinical Knowledge Related to "Cephalohematoma Caused by Birth Injury" | |
|---|---|
| Knowledge Aspect | Corresponding Value |
| 人群
Patient Population | 新生儿
Neonate |
| 好发年龄
Prevalence Ages | 新生儿
Neonate |
| 特发年龄
Onset Ages | 新生儿
Neonate |
| 症状
Primary Symptoms | 头颅血肿;肿胀;疼痛;局部皮色发红;皮肤苍白;贫血;哭闹;食欲减退;惊厥;易激惹;哭声尖;吃奶差;精神萎靡;反应差;昏迷;抽搐;休克;癫痫
Head hematoma; swelling; pain; localized redness of the skin; pallor; anemia; crying; reduced appetite; convulsions; irritability; high-pitched crying; poor feeding; lethargy; poor responsiveness; coma; seizures; shock; epilepsy |
| 伴随症状
Associated Symptoms | 皮肤苍白;贫血;哭闹;食欲减退;惊厥;易激惹;哭声尖;吃奶差;精神萎靡;反应差;昏迷;抽搐;休克;癫痫
Pale skin; anemia; crying; reduced appetite; convulsions; irritability; high-pitched crying; poor feeding; lethargy; poor responsiveness; coma; seizures; shock; epilepsy |
| 鉴别性症状
Differential Symptoms | 头颅血肿;肿胀;疼痛;局部皮色发红
Head hematoma; swelling; pain; localized skin redness |
| 体格检查
Physical Examination | 头部触诊（头颅肿胀 血肿处波动感）
Palpation of the head (feeling for fluctuations at the site of the head swelling hematoma) |
| 解剖部位
Anatomical Sites | 头颅
cranium |
| 影响部位
Affected Sites | 头;脑
Head; brain |
| 人体系统
Affected Body Systems | 皮肤软组织
Cutaneous soft tissues |
| 治疗原则
Treatment Principles | 止血;降颅压;促进脑血流灌注;改善微循环
Hemostasis; intracranial pressure reduction; promotion of cerebral blood flow perfusion; improvement of microcirculation |
| 继发疾病
Secondary Diseases | 癫痫
Epilepsy |
| 药物
Medications | 苯巴比妥或水合氯醛;维生素 K;葡萄糖;甘露醇
Phenobarbital or chloral hydrate; Vitamin K; glucose; mannitol |
| 手术
Surgical Procedures | 颅内血肿清除术
Intracranial hematoma evacuation surgery |
| 辅助检查异常结果
Auxiliary Examinations | 超声检查（观察有无其他损伤）
Ultrasound examination (to observe for any other injuries) |
| 实验室检查异常结果
Laboratory Examinations | 血象检查（红细胞计数 血红蛋白量减少） 血生化检查（间接胆红素增高）
Blood count test (decreased red blood cell count) and blood chemistry test (elevated indirect bilirubin) |
| 科室
Departments | 新生儿科;儿外科
Neonatology; Pediatric Surgery |
| 危重等级
Severity Level | 4 |

Table 15: An illustrative instance showcasing comprehensive knowledge across 18 aspects pertaining to "cephalohematoma caused by birth injury"

# K    EXAMPLES OF LLMS' RESPONSES IN THREE CATEGORIES

| Tier | Disease | Knowledge Aspect | Ground Truth | Original LLM Response | Post-processed LLM Response |
|------|---------|------------------|--------------|----------------------|------------------------------|
| **Completely Wrong** | rheumatoid arthritis of the hand interphalangeal joints | patient population | higher prevalence in females; middle-age; elderly | Ok, I see. | None |
| | wart of brass players | anatomical sites | skin | The anatomical site of wart of brass players includes the bone marrow. | bone marrow |
| | temporal lobe abscess | departments | eurology; neurosurgery | Departments involved in temporal lobe abscess include: None | None |
| **Partially Correct** | tracheobronchial amyloidosis | affected sites | trachea; bronchi; lung | The affected areas of amyloidosis in the bronchi include the lungs and chest. | lung; chest |
| | fascial necrosis | treatment principles | anti-infection; antibacterial; anti-swelling; analgesic; medication | The treatment principles for fascial necrosis include liver protection, gastric protection, antidiarrheal, and pain relief. | liver protection; gastric protection; antidiarrheal; pain relief |
| | punctate urethral opening | laboratory examinations | urinalysis (elevated white blood cell count, visible hematuria upon microscopic examination) | Hello, based on your description, typically, abnormal laboratory test results for a punctate urethral opening include: Complete Blood Count (CBC): Elevated white blood cell count and neutrophil count, with a bone marrow examination showing an increase in immature granulocyte count. | Complete Blood Count (CBC): Elevated white blood cell count and neutrophil count, with a bone marrow examination showing an increase in immature granulocyte count. |
| **Basically Correct** | esophageal abscess | affected body systems | digestive system | Digestive system. | digestive system |
| | humerus greater tuberosity fracture | auxiliary examinations | X-ray Examination (Humerus Greater Tuberosity Fracture). | X-ray Examination (Visible fracture line in the humerus greater tuberosity area on the film). | X-ray Examination (Visible fracture line in the humerus greater tuberosity area on the film). |
| | interferential atrioventricular dissociation | primary symptoms | fatigue; weakness; dizziness; syncope; angina; palpitations; chest tightness; shortness of breath | Common symptoms of interferential atrioventricular dissociation include palpitations; tachycardia or irregular heart rhythm; dizziness; fatigue; chest tightness; shortness of breath; syncope; chest pain; fatigue; weakness. | palpitations; irregular heart rhythm; dizziness; fatigue; chest tightness; shortness of breath; syncope; chest pain; fatigue; weakness |

Table 16: Examples of LLM responses in three tiers.

## L EVALUATION PLATFORM PLANNING

We plan to make our evaluation benchmark freely accessible for any researchers and organizations by releasing an online evaluation platform. The platform provides participants who are seeking to evaluate their LLMs a list of diseases, prompts we employ in this work, and several demonstrative examples. Participants can either choose to directly test their LLMs with the provided prompts, or DIY prompts by themselves. Once the participants upload the LLMs' response on the platform, the evaluation script will be running automatically. The evaluation results, including the performance on various knowledge aspects, will be available for downloading once the evaluation process ends. Participants can choose whether to update their performance on a leaderboard and compare the performance with others.

