# OpenReview forum: "Do Current Large Language Models Master Adequate Clinical Knowledge?"
_ICLR.cc/2024/Conference — Submitted to ICLR 2024_

### Official Review · Reviewer_h42w · 2023-10-31

**Soundness:** 2 fair
**Presentation:** 1 poor
**Contribution:** 2 fair
**Rating:** 3
**Confidence:** 2

**Summary:**

The paper evaluates the current performance of medical LLMs by creating a benchmark and testing existing said LLMs against it. This work creates MedDisK, a database designed to test the medical knowledge of LLMs on different "clinical knowledge aspects". These properties are not limited to those used just for diagnosis; example properties include patient population, treatment principles, departments (relevant medical departments), etc. This work also introduces MedDisKEval, a method that includes automated and clinical-expert-dependent steps to grade the performance of LLMs. Notably, the paper concludes that most current medical LLMs do not perform better than the base LLMs they are built upon.

**Strengths:**

* The development of a medical knowledge benchmark involved consulting 20 clinical experts over 10 months is good. This paper focuses largely on Chinese data/expert consult, but the presentation itself features relevant English translation.
* Creating a clear evaluation method combining automated/expert consultation is also useful.
* The conclusions of the evaluation point out specific flaws in existing medical LLMs; certain models evaluate different features poorly, for example. This provides a concrete criticism/evaluation of those methods that can be built upon.

**Weaknesses:**

* The creation of a medical LLM benchmark itself does not make fundamental improvements over existing benchmarks developed in medical LLM research. As an example, the Singhal et al. 2023a paper also tested modern LLMs with human evaluation (MultiMedQA). Creating another benchmark by itself is not a conceptually novel improvement, and this work did not sufficiently argue for its improvement above these existing models/evaluations.
* This work does not go into as much detail about the representation of medical knowledge in LLMs, providing only a benchmark without technical insight of what the LLMs might be doing or how they encode medical information.
* Using MedDisKEval seems expensive or possibly unreliable. Someone seeking to use this evaluation method may need to consult expert opinion themselves, just to calibrate the alignment scores. The motivation behind the linear combination of BLEU-1, ROUGE-1 and cosine sim is empirically driven and is not inherently convincing as a metric.

**Questions:**

* The database MedDisK was constructed with "clinical experts and machine assistance." Further clarification is required; what was the exact process of constructing the database, and how was machine assistance used?
* This work focused on a set of LLMs that is somewhat disjoint from existing popular medical LLMs. For example, MedPaLM?

---

> ### Author Response · Authors · 2023-11-17
> **Response to Reviewer h42w --- Part 1**
>
> **1. Novelty of MedDisKEval**: Thank you for your thoughtful concern on the novelty of our proposed LLM benchmark. The MultiMedQA dataset proposed by Singhal et al. is actually a set of medical QA datasets, some of them (MMLU, MedQA, MedMCQA, etc.) have already been proposed in other literature. These QA-based datasets are able to evaluate LLMs’ medical capabilities to some extent, while they face limitations in evaluating clinical knowledge mastery. Firstly, existing QA-based evaluation sets have a limited coverage of diseases and disease-related knowledge, which are crucial for clinical practice. To clarify our claim, we have compared them with our proposed benchmark MedDisK in terms of the coverage of diseases, and 7 types of disease-knowledge-related entities: patient population (Popu), symptom (Symp.), body parts (Part.), body systems (Syst.) therapeutic procedure (Proc.), medication (Medi.), and departments (Dept.).We have found that these datasets cover limited number of diseases as well as disease-related knowledge:
>
> | Datasets           |      Type      | \# diseases |    Publicly available?    |
> | ------------------ | :------------: | :---------: | :-----------------------: |
> | MedQA              |   QA dataset   |    1391     |            Yes            |
> | MedMCQA            |   QA dataset   |    3475     |            Yes            |
> | MMLU (medical)     |   QA dataset   |     383     |            Yes            |
> | MedicationQA       |   QA dataset   |     172     |            Yes            |
> | LiveQA             |   QA dataset   |     480     |            Yes            |
> | HealthSearchQA     |   QA dataset   |     262     |            Yes            |
> | **Total of Above** |  QA datasets   |    3907     |            Yes            |
> | **MedDisK (Ours)** | Knowledge base |    10632    | Yes (evaluation platform) |
>
> | Dataset         | \#Popu. | \#Symp. | \#Part. | \#Syst. | \#Proc. | \#Medi. | \#Dept. |
> | --------------- | ------- | ------- | ------- | ------- | ------- | ------- | ------- |
> | MedQA           | 197     | 377     | 574     | 15      | 429     | 62      | 36      |
> | MedMCQA         | 241     | 452     | 1245    | 33      | 811     | 56      | 54      |
> | MMLU  (medical) | 92      | 114     | 250     | 10      | 111     | 6       | 9       |
> | MedicationQA    | 27      | 64      | 52      | 7       | 70      | 67      | 3       |
> | LiveQA          | 75      | 108     | 141     | 9       | 166     | 14      | 19      |
> | HealthSearchQA  | 10      | 63      | 40      | 3       | 4       | 2       | 2       |
> | **Total  of Above** | 349     | 570     | 1362    | 34      | 997     | 183     | 83      |
> | **MedDisK  (Ours)** | 701     | 18737   | 1585    | 89      | 5097    | 3826    | 89      |
>
> The comparison results indicate that the proposed benchmark covers a much broader range of disease knowledge compared to the previous QA datasets. We have also updated this comparison results in the appendix A2 of the revised paper. Moreover, several recent studies ([1], Table 8 in [2]) highlight that **data contamination** in existing LLMs adversely affects the fairness of contemporary open-source QA-based evaluation datasets. In contrast, our knowledge base is crafted by clinical experts and cannot be directly accessed on the Internet; thus, the proposed evaluation benchmark does not suffer from data contamination. An evaluation platform will be released to ensure the availability our proposed benchmark.
>
> [1] Zhou K, Zhu Y, Chen Z, et al. Don't Make Your LLM an Evaluation Benchmark Cheater[J]. arXiv preprint arXiv:2311.01964, 2023.
>
> [2] Wei T, Zhao L, Zhang L, et al. Skywork: A More Open Bilingual Foundation Model[J]. arXiv preprint arXiv:2310.19341, 2023.
>
> **2. Research value**: We are sincerely grateful for your insightful comments. For a considerable period, we have been researching how general LLMs can truly serve as foundational models in the medical domain and be applicable in real clinical scenarios. We have observed that current general/medical LLMs cannot be directly applied in real-world clinical applications, despite achieving considerable performance on specific medical tasks. The motivation of this work is to find out how far current LLMs are to the real medical foundation models in the aspect of clinical knowledge mastery. We humbly believe that the proposed evaluation benchmark could offer valuable insights to developers of medical LLMs, ultimately promoting the advancement of foundational models in the medical domain. We will further delve into how LLMs can better encode medical knowledge to be applicable in real-world clinical scenarios.

---

> > ### Author Response · Authors · 2023-11-17
> > **Response to Reviewer h42w --- Part 2**
> >
> > **3. Cost and Reliability of MedDisKEval**: Thank you so much for your thoughtful concerns of the cost and reliability of the proposed evaluation benchmark.
> >
> > For the cost of MedDisKEval, as we mentioned above, we will make an online evaluation platform freely accessible for any researchers and organizations. The platform provides participants a list of diseases, prompts we employ in this work, and several demonstrative examples. Participants can either directly test their LLMs with the provided prompts, or design prompts by themselves. Once the participants upload the responses of their LLMs on the platform, the evaluation script will be running automatically. The alignment standard is **preset** by our clinical experts in the evaluation script and does not require extra calibration by participants. The evaluation results, including the performance on various knowledge aspects, will be available for downloading once the evaluation process ends. A leaderboard will be available for researchers to compare the performance of their LLMs with others. The planning of this evaluation platform has been updated in the appendix L of our revised paper.
> >
> > For the reliability of MedDisKEval, it is indeed challenging to find proper metrics for our evaluation, and we have not found a perfect metric for our evaluation yet. As a compromise, we utilized common text evaluation metrics, such as BLEU for machine translation, ROUGE for text summarization, and cosine similarity for text similarity calculation. BLEU and ROUGE primarily assess token-level consistency between LLMs' responses and the ground truth context, whereas cosine similarity focuses on semantic-level consistency. We have observed in experiments that token-level metrics face difficulties in dealing with synonyms of medical terms, while cosine similarity may ignore slight token-level difference that completely change the meaning of medical terms. Therefore, we combine these metrics together in our evaluation. It is worth noting that the combination of these metrics is **NOT** achieved through a straightforward linear combination. Instead, each score generated by a specific metric is initially categorized into three tiers using expert-defined thresholds. Considering the statistical distribution biases across metrics, we request experts to establish independent thresholds for each metric. Therefore, the grades derived from different metrics are somewhat comparable and can be combined by **soft voting** to better reflect the performance of LLMs. We have also found that the combined metric achieves high consistency (**0.837**) with human experts (see Table 2 in our paper), indicating the reliability of this metric.

---

> > > ### Author Response · Authors · 2023-11-17
> > > **Response to Reviewer h42w --- Part 3**
> > >
> > > **4. Clarification of MedDisk construction**: We are grateful for your constructive concerns of MedDisK construction. The construction of MedDisK involves two phases: selection of diseases, and knowledge annotation. In the first phase, we conduct a statistical analysis with machine assistance on the occurrence of ICD-10 diseases in ~**4 million** highly de-identified electronic health records (EHRs) from over 100 hospitals across 5 cities. We selected diseases with a frequency exceeding 1/10000, resulting in 1,048 distinct diseases. We further requested experts to select low-frequency diseases that are important in clinical practice from the remaining, resulting in another 9,584 diseases. In the second phase, we employed a retrieve-and-proofread knowledge annotation method. We first exploit an information retrieval module that retrieves disease-related information from medical books and literature. Subsequently, we asked clinical experts to proofread the retrieved information, filtering out irrelevant content, and supplementing missing knowledge. We find that such human-machine collaboration is helpful for minimizing human bias introduced in annotation.
> > >
> > > **5. Evaluated LLMs**: Thank you for your valuable and constructive suggestions. MedPaLM is currently a closed-source LLM that is only available to a selected group of Google Cloud customers for limited testing. Meanwhile, we found that the medical LLMs we evaluated have claimed **considerable** performance on medical QA datasets ([1] [2] [3]), while they are not applicable in real-world clinical scenarios. Therefore, we constructed a disease-based knowledge base, covering over 10k diseases across 18 distinct clinical knowledge aspects, to evaluate the clinical knowledge mastery of current foundation models and medical LLMs. Based on our evaluation results, none of the existing LLMs have mastered adequate clinical knowledge for real clinical applications. In comparison, GPT-3.5-turbo generally exhibits a greater mastery of clinical knowledge than current medical LLMs. We will contact Google in the future to get access of their MedPaLM model and evaluate it on our benchmark.
> > >
> > > [1] Zhang H, Chen J, Jiang F, et al. HuatuoGPT, towards Taming Language Model to Be a Doctor[J]. arXiv preprint arXiv:2305.15075, 2023.
> > >
> > > [2] Wang H, Liu C, Xi N, et al. Huatuo: Tuning llama model with chinese medical knowledge[J]. arXiv preprint arXiv:2304.06975, 2023.
> > >
> > > [3] Yunxiang L, Zihan L, Kai Z, et al. Chatdoctor: A medical chat model fine-tuned on llama model using medical domain knowledge[J]. arXiv preprint arXiv:2303.14070, 2023.

---

### Official Review · Reviewer_tVw5 · 2023-10-31

**Soundness:** 3 good
**Presentation:** 2 fair
**Contribution:** 2 fair
**Rating:** 5
**Confidence:** 4

**Summary:**

To evaluate whether LLMs have mastered sufficient clinical knowledge, the authors first propose a large-scale medical disease-based knowledge base named MedDisK. They then develop MedDisKEval, a method that prompts LLMs to retrieve information on clinical knowledge aspects and measures the similarity between LLM-generated information and MedDisK. Results show that most of the current LLMs do not have sufficient clinical knowledge.

**Strengths:**

1. The motivation is clear, and it is interesting to know whether current LLMs have mastered sufficient domain knowledge to help in the medical domain.
2. The authors conduct extensive experiments with 12 LLMs, which include general LLMs and medical LLMs.
3. The authors provide sufficient examples of prompt instructions, knowledge aspects, and LLM responses, which make it easier for readers to grasp the basic idea of the paper.

**Weaknesses:**

1. One significant issue with this paper is that the authors may overstate the implications of their evaluation results. The experiments are exclusively conducted in Chinese. However, this critical detail is not adequately emphasized in the main paper, particularly in their conclusion that "none of the evaluated LLMs have mastered sufficient knowledge to handle real clinical problems effectively." Based on their evaluation, the valid conclusion should be that LLMs do not possess adequate clinical knowledge **in the Chinese language**, and this finding cannot be generalized to other languages.

2. In the second paragraph of the introduction, the authors claim that current QA-based medical evaluation datasets cannot evaluate whether LLMs have mastered sufficient medical knowledge because those datasets cover only some common diseases. It would be more robust if the authors could further justify this statement with some analysis (e.g. to quantitatively show the coverage of diseases in the existing benchmarks).

3. For the proposed knowledge base MedDisK, it would be better for authors to include more details of the construction process. For example, how is the agreement among the clinical experts, is there any strategy used to tackle disagreement, and will this process introduce any additional human bias?

4. In section 3.2.1, the authors "employ a specialized NER model to identify and extract medical entities from the text". However, the exact name and citation of the used NER model are missing, and it will be more convincing to include an analysis of the accuracy of the NER model as incorrectly recognized entities could impact the evaluation results of LLMs.

**Questions:**

1. The authors claim that current QA-based medical evaluation datasets cover only some common diseases. However, in section 3.1 where the authors introduce their proposed knowledge base, it is said that "We first select a subset from the ICD10 database according to whether the diseases are common in clinical (determined by clinical experts) and are statistically frequent in EHR (Electronic Health Record), resulting in 10,632 common diseases." I wonder why they also consider common diseases in their knowledge base?

2. In the section of "Disease-Knowledge-based Automated Scoring", are there any better metrics to evaluate the similarity? The token-level BLEU-1 and ROUGE-1 cannot consider semantic meaning, and the M3E model is described as a sentence-level metric, whereas the evaluation in this context focuses on the meaning of individual tokens.

3. In Table 3, one completely wrong example of LLM response is "ok, I see". Since the authors mention that they employ a specialized NER model to identify and extract medical entities from the text, I wonder why the NER model could extract such words from the responses.

4. In section 4.2.2, the authors assign scores of 0, 5, and 10 to ”Completely Wrong,” ”Partially Correct,” and ”Basically Correct,” respectively, to calculate a total score. It's important to clarify how they arrived at the values "0, 5, 10" for this scoring system.

---

> ### Author Response · Authors · 2023-11-17
> **Response to Reviewer tVw5 --- Part 1**
>
> **1. Language issue**: Thank you for your thoughtful comments. It is worth noting that foundational medical knowledge is relatively stable and universally applicable, unaffected by specific languages, because it includes fundamental concepts in physiology, anatomy, pharmacology, and other fields. These concepts have similar meanings across different linguistic contexts. Moreover, according to current works on LLM evaluation, foundation models such as GPT-3.5-turbo, GPT-4 also achieves considerable performance on **Chinese** general [1] and medical [2] datasets. This is because current LLMs have already possessed strong multilingual capabilities, and language is no longer a key factor influencing their performance.
>
> We have also conducted a preliminary experiment to study the influence of language in our evaluation. Specifically, we randomly extract 500 diseases in our knowledge base, and translate all the related content into English with GPT-4. We have asked clinical experts to validate the correctness of the translation. Subsequently, we evaluate GPT-3.5-turbo on this small-scale English dataset by replacing the NER model with an English medical NER model MedCAT [3] and M3E model with MPNet model [4]. We find that the total score of GPT-3.5-turbo on these 500 diseases is **4.56** in English and **4.07** in Chinese, indicating that language has limited influence to our proposed evaluation method.
>
> [1] Huang Y, Bai Y, Zhu Z, et al. C-eval: A multi-level multi-discipline chinese evaluation suite for foundation models. Advances in neural information processing systems, 2023.
>
> [2] Zhu W, Wang X, Zheng H, et al. PromptCBLUE: A Chinese Prompt Tuning Benchmark for the Medical Domain[J]. arXiv preprint arXiv:2310.14151, 2023.
>
> [3] Kraljevic Z, Searle T, Shek A, et al. Multi-domain clinical natural language processing with medcat: the medical concept annotation toolkit[J]. Artificial intelligence in medicine, 2021.
>
> [4] Song K, Tan X, Qin T, et al. Mpnet: Masked and permuted pre-training for language understanding[J]. Advances in Neural Information Processing Systems, 2020.
>
> **2. Coverage of existing benchmarks**: We are grateful for your constructive suggestions. We compare MedDisK with six well-known medical QA evaluation datasets: MedQA, MedMCQA, MMLU (medical part), MedicationQA, LiveQA, and HealthSearchQA. We estimate the coverage of diseases in theses benchmarks by leveraging an English medical NER model MedCAT. The results are presented below, and we have added this comparison in the appendix A2 of our revised paper as well:
>
> | Datasets           |      Type      | \# diseases |    Publicly available?    |
> | ------------------ | :------------: | :---------: | :-----------------------: |
> | MedQA              |   QA dataset   |    1391     |            Yes            |
> | MedMCQA            |   QA dataset   |    3475     |            Yes            |
> | MMLU (medical)     |   QA dataset   |     383     |            Yes            |
> | MedicationQA       |   QA dataset   |     172     |            Yes            |
> | LiveQA             |   QA dataset   |     480     |            Yes            |
> | HealthSearchQA     |   QA dataset   |     262     |            Yes            |
> | **Total of Above** |  QA datasets   |    3907     |            Yes            |
> | **MedDisK (Ours)** | Knowledge base |    10632    | Yes (evaluation platform) |
>
> The table above shows that the proposed MedDisK covers much more diseases (>10k) than existing medical QA datasets (~3k). The results suggest that MedDisKEval significantly surpasses existing benchmarks in terms of the disease knowledge coverage.

---

> > ### Author Response · Authors · 2023-11-17
> > **Response to Reviewer tVw5 --- Part 2**
> >
> > **3. Construction details**: Thank you for your insightful concern of MedDisK construction process. The construction of MedDisK can be divided into two phases: disease selection and knowledge annotation. In the first phase, we analyzed the occurrence of ICD-10 diseases in ~**4 million** highly de-identified EHRs from over 100 hospitals across 5 cities, and extracted 1,048 high-frequency diseases. To broaden the coverage of our knowledge base, we engaged clinical experts to identify another 9,584 clinically significant diseases from the remaining low-frequency diseases. As a result, we select a total of 10,632 diseases.
> >
> > In the knowledge annotation phase of MedDisK construction, each clinical expert is responsible for a specific type of diseases. The introduction of human bias in this process is minimal, given that medical knowledge is objective information extensively recorded in a vast amount of medical literature and books. To minimize the human bias introduced in labeling, we use a retrieval module to retrieve related information of the specific disease from medical books and literature. We then asked experts to proofread the retrieved information and craft each clinical knowledge aspect of the given disease. If an expert encountered uncertainty regarding certain content, they would report the issue, and a discussion among all experts would arrive at a consensus for the annotation result. To test the feasibility of this annotation approach, in the early stage of knowledge base construction, we engaged two experts to annotate the knowledge related to 20 diseases (approximately involving **1,000** disease-related knowledge points) using the human-machine collaborative method mentioned above. The results revealed a disagreement rate of less than 2%, demonstrating the reliability of our knowledge base construction method. We have updated these construction details of MedDisK in appendix A1 of the revised paper.
> >
> > **4. NER model**: We are grateful for your constructive and thoughtful suggestions on the introduction of our medical NER model. The utilization of NER model in our evaluation can filter out irrelevant content in LLM responses and improve the reliability of the evaluation. In our study, we employ a medical NER model that has been trained on numerous EHRs early and applied in various real-world medical scenarios, such as assisted consultations and diagnosis, EHR-based semantic parsing. Specifically, we first pre-trained a BERT-based model on 3.5 million highly de-identified EHRs from 7 hospitals with MLM objective. Then we finetuned the pre-trained model on a total of 200k labeled EHR segments following the method proposed in [1]. On a test set of 10k+ real-word EHRs involving 40k+ medical entities, our NER model achieves a micro-f1 score of **0.88** on a total of **116** types of medical entities, and the f1-score even surpasses 0.9 on some common medical entities, such as anatomical sites, symptoms, medication. We have provided more details of our NER model in the appendix E of the revised paper.
> >
> > [1] Jianlin Su, Ahmed Murtadha, Shengfeng Pan, Jing Hou, Jun Sun, Wanwei Huang, Bo Wen, and Yunfeng Liu. Global pointer: Novel efficient span-based approach for named entity recognition. arXiv preprint arXiv:2208.03054, 2022.
> >
> > **5. About Common disease**: Thank you for carefully reading our paper. The 10,632 “Common Diseases” in our MedDisK refer to a subset relative to the total 27,000 ICD-10 diseases. We find in our preliminary study (Table 7 in the revised paper) that existing QA-based medical evaluation datasets cover limited diseases as well as not possess sufficient, systematic disease-centered clinical knowledge. Moreover, according to existing studies ([1], Table 8 in [2]), existing QA-based datasets suffer from data contamination that is widely existed in current LLMs. Motivated by these issues, we constructed a large-scale disease-centered knowledge base to evaluate whether LLMs master adequate clinical knowledge for real-world medical scenarios, and to indicate possible pathways for training medical foundation models. While MedDisK only contains “common diseases” from the ICD-10 disease base, it already includes a substantially greater number of diseases compared to existing QA datasets.
> >
> > [1] Zhou K, Zhu Y, Chen Z, et al. Don't Make Your LLM an Evaluation Benchmark Cheater[J]. arXiv preprint arXiv:2311.01964, 2023.
> >
> > [2] Wei T, Zhao L, Zhang L, et al. Skywork: A More Open Bilingual Foundation Model[J]. arXiv preprint arXiv:2310.19341, 2023.

---

> > > ### Author Response · Authors · 2023-11-17
> > > **Response to Reviewer tVw5 --- Part 3**
> > >
> > > **6. Metrics**: Thank you for your constructive and insightful suggestion. Measuring the correctness and completeness of LLMs’ response is considered as a challenge in NLP community. To ensure the fairness and robustness of our evaluation, we employ multiple metrics, including BLEU, ROUGE, and cosine similarity. BLEU and ROUGE are strict metrics that consider the token-level consistency, while cosine similarity focuses on semantic-level consistency. It is worth noting that LLMs’ responses have already been parsed into segments for enumerated types of knowledge, and M3E is acted as an embedding model rather than a sentence encoder in this situation. We have tried other token-level semantic similarity metrics such as **BERTScore** [1] in our study and found that it achieves unsatisfied consistency with manual evaluation (see **Table 11** in the appendix of the revised paper). We hypothesize that the reason BERTScore do not work in our evaluation is because in the medical domain, slight differences in tokens between terms may lead to entirely different meanings. For example, “acute renal failure” and “acute respiratory failure” are completely different diseases, where there is only a slight difference in tokens.
> > >
> > > [1] Zhang T, Kishore V, Wu F, et al. BERTScore: Evaluating Text Generation with BERT[C]//International Conference on Learning Representations. 2019.
> > >
> > > **7. Table 3 content**: We sincerely appreciate your carefully reading. The LLM response provided in Table 3 is the original response of LLMs before sending into the NER model. Based on your valuable suggestions, we have revised our paper by adding more examples with both original and post-processed responses in **Table 16** of appendix K.
> > >
> > > **8. Scoring System**: Thank you. The scoring system is derived based on our observation of LLMs’ response. We observed that in the responses of a certain LLM, some are unrelated to the disease, some partially overlap with our annotations, and the rest responses basically cover all the knowledge stored in our knowledge base. For the convenience of evaluation, we established a scoring system on a scale of 0 to 10, where 0 represents “completely wrong”, 5 indicates “partial correct”, and 10 signifies “basically correct”.

---

### Official Review · Reviewer_rk49 · 2023-11-03

**Soundness:** 2 fair
**Presentation:** 3 good
**Contribution:** 2 fair
**Rating:** 5
**Confidence:** 4

**Summary:**

The paper introduces a large-scale medical disease-based knowledge base MedDisK, covering 10,632 common diseases and 18 clinical knowledge to evaluate LLMs. The purpose of the dataset is to  (a) include common diseases (b) involve disease base knowledge and (c) ensure that the sourcing of the dataset is such that it remains publicly inaccessible to prevent leaks during testing.

First filter common diseases (determined by clinical experts based on ICD10 databases and frequency in EHR)  resulting in 10,632 common diseases. Then  employ clinical experts to define 18 disease-based clinical knowledge aspects that are crucial to medical decision-making (diagnoses, examinations, treatments) for each of the diseases. They use this database to probe LLMs and evaluate the mastery of clinical knowledge. They show that their scoring measures are in high agreement with clinical experts' subjective evaluation.

Using the evaluation metrics they show that existing LLMs have not mastered adequate knowledge for clinical practice (showing that over 50%  of the generated information is not consistent with their KB) and are not ready to be foundation models for clinical domain.

**Strengths:**

The paper does a good job in communicating the ideas. I agree with the author's motivation that for the LLMs to be accepted as foundation models they need to have mastered adequate clinical knowledge. This is an important question and needs comprehensive evaluation.

**Weaknesses:**

The paper could provide a more thorough justification for the introduction of the new medical dataset, especially in the context of existing evaluation datasets. The paper mentions that the existing evaluation datasets cover only some common diseases and lack extensive coverage of knowledge across various aspects of diseases. This reviewer feels that this needs to be substantiated with more thorough comparison.

While it is surprising that most of the LLMs perform poorly (with over 50%) predicted to be completely wrong. The evaluation procedure used to arrive at this conclusion requires further elaboration.

Overall I am not fully convinced that this dataset MedDisK  and the outlined evaluation procedure is robust for determining LLMs clinical knowledge yet.

This reviewer has listed all the concerning questions in detail below.

**Questions:**

What is the source of the EHR resource used in the preliminary making of the dataset?

The authors state “The existing medical evaluation benchmarks are predominantly based on question-answering (QA) tasks. These benchmarks collect questions from diverse sources, including medical examinations, electronic health records, online resources, and expert crafting……cover only some common diseases and lack extensive coverage of knowledge across various aspects of diseases. ”  Can you compare each of these resources the paper is referring to in this sentence with MedDisK in terms of coverage of common diseases, disease base knowledge and public availability?  How does this compare with other existing medical relational databases - MIMIC, i2b2, iBKH KG etc?

I understand that the paper does interval sampling (10 examples from each interval) and engages clinical experts to provide a categorical standard - wrong, correct or partially correct.  And this resulted in the following standard (0-0.3 is wrong) and (0.3 to 0.8 is partially correct) and (0.8 to 1.0 is correct). How representative are these categories? Did the experts find that all the samples in 0.8 to 1.0 are correct and correspondingly all in 0-0.3 are wrong? Can you provide more representative examples or more thorough classification of the “Completely Wrong” category?

Since LLMs response is post-processed using the NER model I think the NER model's performance is extremely crucial to evaluation. How well does it perform in identifying medical entities? From the analysis conducted in Table 8, it appears that all the LLMs are underperforming in identifying symptoms, affected sites, etc., while they generally perform well in recognizing population ages involving numeric entities.

Would it be considered a correct hit if the model predicts 'GI tract' instead of 'digestive system' in the examples from Table 3? What kind of standardization was performed in evaluating LLMs response with the experts output?

What according to the authors are the limitations of the dataset and the evaluation procedure outlined here?

---

> ### Author Response · Authors · 2023-11-17
> **Response to Reviewer rk49 --- Part 1**
>
> **1. About MedDisK construction**: Thank you for your thoughtful concerns on the construction of our proposed database. As we introduced in our paper, the ICD-10 diseases coding system includes around 27,000 diseases. First, we conducted a statistical analysis of the occurrence frequencies of these ICD-10 diseases based on ~**4 million** highly de-identified EHRs from > 100 hospitals across 5 cities. We then filtered out high-frequency diseases with an occurrence rate exceeding 1/10000 to build the disease knowledge base. This process results in 1,048 diseases. To expand MedDisK's coverage of diseases, we asked clinical experts to choose an additional 9,584 diseases from the remaining low-frequency cases based on their clinical significance. We have included additional construction details of MedDisK in appendix A1 of the revised paper.
>
> **2. MedDisKEval versus Other datasets**: We are sincerely grateful to your constructive comments on the comparison between MedDisKEval and other medical QA datasets. We choose a total of six medical QA datasets for comparison: MedQA, MedMCQA, MMLU (medical part), MedicationQA, LiveQA, and HealthSearchQA. We use an English medical NER tool called MedCAT [1] to extract diseases from the questions, options, and answers. In assessing disease-based knowledge coverage, we find it challenging to precisely quantify the number of knowledge aspects covered by the QA dataset, given its potential to encompass distinct types of knowledge for various diseases. For instance, these QA datasets may only examine symptoms of one disease and anatomical sites of another. Instead, we extract and analyze the occurrence of 7 common disease-knowledge-related entities within these datasets, including patient population (Popu), symptom (Symp.), body parts (Part.), body systems (Syst.) therapeutic procedure (Proc.), medication (Medi.), and departments (Dept.). The experimental results are presented below, respectively. We have also updated the results and analysis in appendix A2 of the revised paper.
> | Datasets           |      Type      | \# diseases |    Publicly available?    |
> | ------------------ | :------------: | :---------: | :-----------------------: |
> | MedQA              |   QA dataset   |    1391     |            Yes            |
> | MedMCQA            |   QA dataset   |    3475     |            Yes            |
> | MMLU (medical)     |   QA dataset   |     383     |            Yes            |
> | MedicationQA       |   QA dataset   |     172     |            Yes            |
> | LiveQA             |   QA dataset   |     480     |            Yes            |
> | HealthSearchQA     |   QA dataset   |     262     |            Yes            |
> | **Total of Above** |  QA datasets   |    3907     |            Yes            |
> | **MedDisK (Ours)** | Knowledge base |    10632    | Yes (evaluation platform) |
>
> | Dataset         | \#Popu. | \#Symp. | \#Part. | \#Syst. | \#Proc. | \#Medi. | \#Dept. |
> | --------------- | ------- | ------- | ------- | ------- | ------- | ------- | ------- |
> | MedQA           | 197     | 377     | 574     | 15      | 429     | 62      | 36      |
> | MedMCQA         | 241     | 452     | 1245    | 33      | 811     | 56      | 54      |
> | MMLU  (medical) | 92      | 114     | 250     | 10      | 111     | 6       | 9       |
> | MedicationQA    | 27      | 64      | 52      | 7       | 70      | 67      | 3       |
> | LiveQA          | 75      | 108     | 141     | 9       | 166     | 14      | 19      |
> | HealthSearchQA  | 10      | 63      | 40      | 3       | 4       | 2       | 2       |
> | **Total  of Above** | 349     | 570     | 1362    | 34      | 997     | 183     | 83      |
> | **MedDisK  (Ours)** | 701     | 18737   | 1585    | 89      | 5097    | 3826    | 89      |
>
> The experimental results show that MedDisK covers significantly more diseases and disease-knowledge-related entities than existing QA datasets, which may demonstrate our claim. For medical knowledge graphs like iBKH, they contain various biomedical entities (disease, gene, molecule, etc.) and their corresponding relations. Generally, these medical KGs cover only a limited number of disease-related relations; for instance, iBKH includes only 6 disease-related relations. In contrast, MedDisK is a disease-centered knowledge base that involves **18** essential knowledge types crucial for clinical practice, including diagnosis and treatment. Additionally, none of these medical knowledge graphs have been utilized in the construction of mainstream medical evaluation datasets yet, making them incomparable to our evaluation dataset.
>
> [1] Kraljevic Z, Searle T, Shek A, et al. Multi-domain clinical natural language processing with medcat: the medical concept annotation toolkit[J]. Artificial intelligence in medicine, 2021, 117: 102083.

---

> > ### Author Response · Authors · 2023-11-17
> > **Response to Reviewer rk49 --- Part 2**
> >
> > **3. Results Categories**: Thank you for your insightful comments. We have provided more examples of “Completely Wrong”, “Partially Correct”, and “Basically Correct” in Appendix K of the revised paper. Here are some representative examples of the “Completely Wrong” category:
> > |                         Disease                         |  Knowledge  Aspect  |                    Ground Truth                    |                    Original LLM  Response                    | Post-processed  Response |
> > | :-----------------------------------------------------: | :-----------------: | :------------------------------------------------: | :----------------------------------------------------------: | :--------------------------: |
> > | rheumatoid arthritis of the hand interphalangeal joints | patient  population | higher  prevalence in females; middle-age; elderly |                         Ok,  I see.                          |             None             |
> > |                  wart of brass players                  |  anatomical  sites  |                        skin                        | The  anatomical site of wart of brass players includes the bone marrow. |         bone  marrow         |
> > |                  temporal lobe abscess                  |     departments     |              eurology;  neurosurgery               | Departments  involved in temporal lobe abscess include: None |             None             |
> >
> > Currently, our threshold-based grading strategy is not 100% accurate. However, according to the results of human evaluation on 150 samples, only a small fraction of responses in the “Basically Correct” category contain knowledge errors, and these responses typically had only one or two errors. Similarly, in the “Completely Wrong” category, only a very small portion of responses include correct knowledge. Moreover, the consistency between our proposed automated grading and expert grading reached **0.837** (Table 2 in our paper), indicating a strong agreement between automated and clinical expert evaluation. Therefore, the categorization of results is generally reasonable. We are greatly appreciated your valuable comments, and we plan to enhance the accuracy of this categorization through a more complex post-processing module in the future.
> >
> > **4. NER model**: We would like to express gratitude for your insightful thoughts and valuable suggestions of our medical NER model. The Medical NER model is indeed a crucial module in our evaluation method. In fact, NER models also play important roles in various medical applications. The NER model we applied in our evaluation is a reliable and stable tool that has been applied in various medical scenarios, including assisted diagnosis, EHR-based semantic parsing, and assisted consultations. It is a BERT-based model that is constructed by first pretraining with MLM objective on 3.5 million unlabeled and highly de-identified EHRs from 7 hospitals, and finetuning on 200k labeled EHR segments following the method proposed in [1]. The model is able to identify a total of **116** types of important medical concepts, and it achieves a micro-f1 of **0.88** on a test set of 10k+ real-world EHRs containing 40k+ medical entities, and even achieves f1 scores exceeding 0.9 on several crucial medical entities (anatomical sites, symptoms, medication, etc.). We also update the details of this NER model in the appendix E of the revised paper.
> >
> > For the performance of LLMs on different knowledge aspects, we conduct an extra analysis of LLMs performance across 18 knowledge aspects, where we find that LLMs perform distinctly on different aspects. We have updated the detailed performance of LLMs on various aspects in **Figure 9 and 10** of the revised paper, respectively. Table 8 (Table 14 in the revised paper) presents the relative performance of 6 medical LLMs compared with their base models. Positive values (in green) indicate the medical LLM outperforms its base model in the corresponding knowledge aspect, while negative values (in red) signify underperformance in the corresponding aspect.
> >
> > [1] Jianlin Su, Ahmed Murtadha, Shengfeng Pan, Jing Hou, Jun Sun, Wanwei Huang, Bo Wen, and Yunfeng Liu. Global pointer: Novel efficient span-based approach for named entity recognition. arXiv preprint arXiv:2208.03054, 2022.

---

> > > ### Author Response · Authors · 2023-11-17
> > > **Response to Reviewer rk49 --- Part 3**
> > >
> > > **5. Standardization**: Thank you for your valuable comments. We have standardized LLMs’ response on a few knowledge aspects (Departments, Ages) with our medical thesaurus, while we also observe that certain medical terms are fundamentally distinct despite their semantic similarities. For example, though “GI tract” and “digestive system” are semantically similar, they are distinguished in the medical field, since the GI tract is a subset of the digestive system, focusing on the direct passage of food.
> > >
> > > It is widely recognized that distinguishing whether two medical terms have entirely identical meanings is a challenge in the field of medical NLP. To address this, we employ multiple metrics in our evaluation. BLEU and ROUGE are strict metrics, emphasizing token-level consistency, while cosine similarity focuses on semantic-level consistency. Our experiments reveal that utilizing multiple metrics ensures the robustness and fairness of the evaluation results.
> > >
> > > **6. Limitations**: Thank you for your concern on the limitations of our evaluation. There are two main limitations of our evaluation dataset:
> > >
> > > 1. Although MedDisK covers a significantly larger number of diseases and disease-based knowledge, it has not covered all the 27,000 diseases in ICD-10 yet. Currently, our evaluation approach cannot assess LLMs’ mastery on some rare diseases in clinical practice.
> > >
> > > 2. MedDisK focuses on knowledge aspects that are crucial for clinical practice, such as medication, examination, surgical procedures. It does not involve other biomedical knowledge, such as molecule structure and gene.

---

### Meta-Review · Area_Chair_VY8u · 2023-12-06

**Metareview:**

This paper introduces a medical knowledge base intended to be use to probe the implicit medical "knowledge" in LLMs. Specifically, the authors introduce "MedDisK", which is constructed on the basis of ICD codes; the authors identify 10,632 prevalent diseases and enlist domain experts to provide definitions for key clinical aspects relevant to these. This is used a probe: The same information is elicited from LLMs, and the resultant output is evaluated first by experts, and then automatically as a function of correlative measures such as ROUGE. It should be noted that this evaluation is done in Chinese.

The results—namely that for 50% of rows in MedDisK, LLMs are completely wrong—are surprising (and interesting), if taken at face value. The dataset may be a useful resource for researchers in biomedical NLP going forward. That said, reviewers largely agreed that the setup here was not particularly well motivated; why are these "clinical knowledge aspects" critical? And does it matter if the models are capable of performing specific "downstream" tasks (e.g., extraction) well, as has been empirically demonstrated? The authors assert, but without much argument, that LLMs must "master adequate knowledge" to be used in this space, but this is not clear to me a priori.

Further, as pointed out by rk49, the procedure being used here may not be robust; for instance, some of the "completely wrong" LLM responses at least appear to be an issue with the prompt strategy. For example, responding "ok, I see" in Table 3 indicates a model has "misunderstood" the question; perhaps this is part of the analysis, but again it seems unlikely that one would explicitly probe for these attributes in practice (i.e., outside of an endeavor to probe implicit knowledge).

Still, I do think critical analyses such as this are important, and I think if the authors can better motivate the intent of the MedDisK dataset and justify its construction (with an eye on downstream tasks especially), and provide further evidence concerning the validity of the measure being used here, this could be a nice contribution.

**Justification For Why Not Higher Score:**

While the proposed resource (and accompanying evaluation) is intriguing, the work would benefit from a more explicit framing of its aims (including making the case that LLMs ought to "master" this particular type of "knowledge" in order to be useful for medical language processing tasks), and further evidence supporting the validity of the automated assessment procedure used.

**Justification For Why Not Lower Score:**

N/A

---

### Decision · Program_Chairs · 2024-01-16

Reject